# PAC: Assisted Value Factorisation with Counterfactual Predictions in Multi-Agent Reinforcement Learning

**Hanhan Zhou**
The George Washington University
hanhan@gwu.edu

**Tian Lan**
The George Washington University
tlan@gwu.edu

**Vaneet Aggarwal**
Purdue University
vaneet@purdue.edu

## Abstract

Multi-agent reinforcement learning (MARL) has witnessed significant progress with the development of value function factorization methods. It allows optimizing a joint action-value function through the maximization of factorized per-agent utilities. In this paper, we show that in partially observable MARL problems, an agent's ordering over its own actions could impose concurrent constraints (across different states) on the representable function class, causing significant estimation errors during training. We tackle this limitation and propose PAC, a new framework leveraging Assistive information generated from Counterfactual Predictions of optimal joint action selection, which enable explicit assistance to value function factorization through a novel counterfactual loss. A variational inference-based information encoding method is developed to collect and encode the counterfactual predictions from an estimated baseline. To enable decentralized execution, we also derive factorized per-agent policies inspired by a maximum-entropy MARL framework. We evaluate the proposed PAC on multi-agent predator-prey and a set of StarCraft II micromanagement tasks. Empirical results demonstrate improved results of PAC over state-of-the-art value-based and policy-based multi-agent reinforcement learning algorithms on all benchmarks.

## 1 Introduction

Many real-world reinforcement learning (RL) problems, such as autonomous vehicle coordination [1] and network packet delivery [2], often involve coordination among multiple entities and are naturally formulated as multi-agent reinforcement learning (MARL). Factorization-based methods have greatly progressed in dealing with the exponentially growing joint state-action space in MARL. Under the notion of Centralized Training and Decentralized Execution (CTDE), algorithms like VDN [3] and QMIX [4] learn a centralized joint action-value function $Q_{tot}$ through a monotonic factorization into local per-agent value functions so that $Q_{tot}$ can be maximized as long as each per-agent value function is maximized by local action selection. Even conditioned on joint state information, a monotonic mixing network for $Q_{tot}$ is shown to restrict the representable function class. Despite efforts to mitigate this, e.g., QTRAN [5] and WQMIX [6], in practice, they empirically perform poorly in complex MARL environments with partial observability [7, 8, 9].

In this paper, we show that in partially observable MARL problems (as exemplified by a multi-state matrix game), an agent's ordering over its own actions could impose concurrent constraints on the representable action-value $Q_{tot}$ in different states. This restriction causes large estimation errors of $Q_{tot}$ during training. It cannot be addressed by existing methods, e.g., adding a state-value correction

term (like QTRAN [5]) or introducing importance weights on dominant state-actions (like WQMIX [6]). It aggravates the relative over-generalization problem [10] – when fully decomposed, per-agent value functions only depend on partial observations and local actions. It renders optimal decentralized policies unlearnable when the employed value function does not have enough representational ability. Solving tasks that require significant coordination in partially observable problems remains a key challenge.

The key insight of this paper is that an accurate factorization in partially observable MARL problems requires improved representation of the value functions, which is crucial to supporting the learning of optimal decentralized policies. We propose a novel architecture, denoted as PAC, for assisted value factorization with counterfactual predictions. It leverages a counterfactual baseline that marginalizes out an agent's potential optimal action while keeping all other agents' actions fixed. The counterfactual predictions of potential optimal actions enable (i) training assistive information that is generated using the variational inference method to expand the representational ability of value functions, and (ii) directly optimizing the factorization of $Q_{tot}$ through a new counterfactual loss. We note that in contrast to communication-based methods like NDQ [11], the use of assistive information in PAC aims to directly improve per-agent value functions in factorization. Minimizing our proposed counterfactual loss together with an information bottleneck loss ensures that such assistive information is relevant, optimal, and succinct for value function factorization in PAC.

Relying on the accuracy of assisted factorization, we further decouple the decision-making of local value function (through separate policy networks) from value function networks, which allows PAC to maintain full CTDE despite the use of assistive information during training and also enables the maximization of entropy to encourage exploration. The accuracy of assisted factorization makes PAC outperform policy-based methods with factorization like DOP [12] and FOP [13]), especially on difficult tasks that require more coordination among the agents since sub-optimality from one agent's policy might propagate and aggravate the training of other agents through the centralized critic [12, 14]. Our key contributions are summarized as follows:

- We propose a novel method, PAC, the first framework for value function factorization by providing variational-encoded counterfactual predictions as assistive information to facilitate per-agent value function estimation.
- The counterfactual predictions can be efficiently computed from a feed-forward baseline $Q^*$ and based on a local search to facilitate direct optimization of $Q_{tot}$ factorization through a new counterfactual loss.
- PAC decouples individual agents' policy networks from value function networks and thus maintains fully decentralized execution while enjoying the benefits of assisted value function factorization. It also leads to an entropy maximization MARL for more effective exploration.
- We demonstrate the effectiveness of PAC and show that PAC significantly outperforms both state-of-the-art value-based and policy-based multi-agent reinforcement learning algorithms on the StarCraft II micromanagement challenge in terms of better performance and faster convergence.

## 2 Model and Background

**Model:** Consider a fully cooperative multi-agent task as decentralized partially observable Markov decision process (DEC-POMDP) [15], given by a tuple $G = \langle I, S, U, P, r, Z, O, n, \gamma \rangle$, where $I \equiv \{1, 2, \cdots, n\}$ is the finite set of agents. The state is given as $s \in S$, from which each agent draws its own observation from the observation function $o_i \in O(s, i) : S \times A \rightarrow O$. At each timestamp $t$, each agent $i$ choose an action $u_i \in U$, composing a joint action selection $\boldsymbol{u}$. A shared reward is then given as $r = R(s, \boldsymbol{a}) : S \times \mathbf{U} \rightarrow \mathbb{R}$, with the next state of each agent is $s'$ with transition probability function $P(s'|s, \mathbf{u}) : S \times U \rightarrow [0, 1]$. Each agent has an action-observation history $\tau_i \in \mathrm{T} \equiv (O \times U)^*$ from its limited local observations. Then the overall objective is to find a joint policy $\boldsymbol{\pi} = \langle \pi_1, ..., \pi_n \rangle$ which corresponds to the joint action-value function $Q(s_t, \boldsymbol{u}_t) = \mathbb{E}[R_t|s_t, \boldsymbol{u}_t]$, that is used to maximize the joint policy function $V(\boldsymbol{\tau}, \boldsymbol{u}) = \mathbb{E}_{s_{0:\infty}, \boldsymbol{u}_{0:\infty}} [\sum_{t=0}^{\infty} \gamma^t r_t | s_0 = s, \boldsymbol{u}_0 = \boldsymbol{u}, \boldsymbol{\pi}]$, and $\gamma \in [0, 1)$ is the discount factor. Quantities in bold denote a joint quantities across all agents, and quantity with super script $i$ denote a quantity specifically belong to agent $i$.

**Value Decomposition:** Following the paradigm of CTDE, VDN [3] and QMIX [4] are popular and representative methods for value function decomposition which learn a centralized action value $Q\text{tot}$ through value decomposition assuming its additivity and monotonicity. In VDN, per-agent sum of the local value is used to calculate the action value $Q^{\text{tot}} = \sum_{a=1}^{n} Q_a(\tau_a, u_a)$. In

QMIX, a monotonic mixing function of each agents' local utilities is proposed as $Q^{\text{tot}}(\boldsymbol{\tau}, \mathbf{u}, s; \theta) = f_\theta\left(\boldsymbol{s}, [Q_1(\tau_1, a_1), ..., Q_n(\tau_n, a_n)]\right)$, where $\frac{\partial Q^{\text{tot}}(\boldsymbol{\tau}, \boldsymbol{u})}{\partial Q_i(\tau_i, u_i)} > 0, \forall i \in \mathcal{N}$. The monotonic mixing function is able to ensure the global optimal $Q^{\text{tot}}$ yields the same result as the set of individual optimal $Q_i$, where $f_\theta$ is approximated by the monotonic mixing network parameterized by $\theta$. The weights are generated by a separate hyper-network that conditions on the global state, where its monotonicity is ensured by performing absolution function on generated parameters for non-negative mixing weights. Then QMIX is trained in a way like DQN [3] with goal to minimise the squared TD error on minibatch of $b$ samples from the replay buffer as $\sum_{i=1}^{b}(Q^{\text{tot}}(\boldsymbol{\tau}, \mathbf{u}, s; \theta) - y^{\text{tot}})^2$ , where $y^{\text{tot}} = r + \gamma \max_{u'} Q^{\text{tot}}(\boldsymbol{\tau'}, \mathbf{u'}, s'; \theta')$, $r$ is the global reward and $\theta'$ is the parameters of the target network whose parameters are periodically copied from $\theta$ for training stabilization.

# 3  Existing Limitations in Partially Observable Multi-state Problems

**(a) Payoff of 2-state matrix game**

| State 1 | | | | State 2 | | | |
|---|---|---|---|---|---|---|---|
| $u_1$\\$u_2$ | A | B | C | $u_1$\\$u_2$ | A | B | C |
| A | 4 | -2 | -2 | A | -2 | 0 | 0 |
| B | -2 | 0 | 0 | B | 4 | -2 | -2 |
| C | -2 | 0 | 0 | C | -2 | 0 | 0 |

**(b) QMIX: $Q_1$, $Q_2$, $Q_{tot}$**

| State 1 | | | | State 2 | | | |
|---|---|---|---|---|---|---|---|
| $Q_1$\\$Q_2$ | 0.3 | -1.2 | -2.6 | $Q_1$\\$Q_2$ | 0.4 | -1.3 | -2.2 |
| -0.2 | 0.1 | -1.0 | -1.0 | -0.2 | 0.4 | -1.0 | -1.0 |
| 0.3 | 1.1 | -0.9 | -1.0 | 0.3 | 1.4 | -0.9 | -1.0 |
| -2.6 | -1.0 | -1.0 | -1.0 | -2.6 | -1.0 | -1.0 | -1.0 |

**(c) WQMIX: $Q_1$, $Q_2$, $Q_{tot}$**

| State 1 | | | | State 2 | | | |
|---|---|---|---|---|---|---|---|
| $Q_1$\\$Q_2$ | 1.0 | 0.2 | 0.2 | $Q_1$\\$Q_2$ | 0.0 | 0.5 | 0.5 |
| 1.0 | 4.0 | -1.2 | -1.2 | 1.0 | -1.2 | 0.1 | 0.1 |
| 0.0 | -0.2 | -1.6 | -1.6 | 0.0 | -1.6 | -1.2 | -1.2 |
| 0.0 | -0.2 | -1.6 | -1.6 | 0.0 | -1.6 | -1.2 | -1.2 |

**(d) PAC (ours): $Q_1$, $Q_2$, $Q_{tot}$**

| State 1 | | | | State 2 | | | |
|---|---|---|---|---|---|---|---|
| $Q_1$\\$Q_2$ | 0.7 | -2.0 | -2.3 | $Q_1$\\$Q_2$ | 0.6 | -2.1 | -2.0 |
| 0.7 | 4.0 | -2.1 | -2.1 | -1.2 | -1.8 | -2.5 | -2.5 |
| -2.0 | -2.1 | -2.4 | -2.4 | 1.6 | 4.0 | -2.0 | -2.0 |
| -2.1 | -2.1 | -2.4 | -2.4 | -1.8 | -2.1 | -2.5 | -2.5 |

Figure 1: In partially observable multi-state games, $Q_{tot}$ is limited to a restricted function class imposing concurrent inequality constraints on $Q_{tot}$ in different states. It causes large factorization error and thus erroneous computation of $\text{argmax}\,(Q_{tot})$. PAC successfully addresses this problem by leveraging assistive information trained using counterfactual predictions, with the direct goal of achieving better value function factorization.

The limitations of monotonic value function factorization have been identified via single-state matrix games, which inspired new algorithms like QTRAN [5] and Weighted-QMIX [6]. In this paper, we show in multi-state problems with partial observability, that one agent's ordering over its own actions could impose concurrent inequality constraints on the joint action-value $Q_{tot}$ in different states, resulting in restrictive function representations of $Q_{tot}$ with large estimate error.

Consider a Markov decision process (MDP) consisting of 2 states with 0.5 transition probabilities between them and two payoff matrices shown in Figure 1(a). Suppose that agent 1 has the same partial observation $o_1$ in states $s^{(1)}$ and $s^{(2)}$. Then, its per-agent value function $q_1(\cdot, \tau_1)$ computed from partial observation $o_1$ are also the same in both states. Due to the monotonicity of the mixing network (even though it is provided with complete joint state information), for any $u_1$ and $u_1'$ with ordering $q_1(u_1, \tau_1) \geq q_1(u_1', \tau_1)$ without loss of generality, we must simultaneously have $Q_{tot}(u_1, u_2, s^{(1)}) \geq Q_{tot}(u_1', u_2, s^{(1)})$ and $Q_{tot}(u_1, u_2, s^{(2)}) \geq Q_{tot}(u_1', u_2, s^{(2)})$ for any action $u_2$ of agent 2 in both states. Representing $Q_{tot}$ on this restricted function class results in significant error in QMIX as shown in Figure 1(b). Although Weighted-QMIX introduces an importance weighting on the dominant state-actions of this game, it only improves the approximation in state 1 and yet causes even higher error in state 2 (in Figure 1(c)). This is exactly because of the inequality restrictions simultaneously imposed on both states, limiting the representational ability of the value functions in partially observable problems.

Clearly, additional information is needed to facilitate successful factorization in these partially observable multi-state problems. It is also worth noticing that even with agent-wise communications, NDQ [11] also fails the task since the communication messages and related loss functions are not designed to drive better factorization. It underscores the importance of making effective use of the right information for successful factorization. We put the results from other methods in Appendix A.3. Our proposed PAC (in Figure 1(d)) addresses this issue by leveraging assistive information trained by a novel notion of counterfactual predictions. More precisely, counterfactual predictions of potentially

optimal agent actions are readily computed from an unrestricted, feed-forward baseline $Q^*$. Training per-agent value functions using a new counterfactual assistance loss leads to substantially improved estimate $Q_{tot}$ and correct $(\arg\max Q_{tot})$ in different states. We compare PAC with a number of state-of-the-art value-based and policy-based methods with function factorizations and demonstrate its performance in challenging partially observable tasks that require significant coordination.

## 4  Methods

To overcome the limitations of existing Value Decomposition methods discussed in Sec. 3, in this section, we introduce the idea of assisted optimal joint policy factorization and propose such a method under the multi-agent soft actor-critic framework.

**Framework overview.** Fig. 2 shows the architecture of the learning framework. There are three main components in PAC : (1) a weighted $Q^{tot}$ utilizing per-agent local critics $q_i(u_i, \tau_i, m_i)$, where $m_i$ serves as assistive information aiding value factorization, (2) an unrestricted joint action estimator $\hat{Q}^*$, which serves as a baseline estimator of $Q^*$ and allows the computation of counterfactual predictions from a quick local search, and (3) an assisted information generating module, which is able to utilize the deep variational bottleneck method to encode the counterfactual optimal joint action selection. In addition to TD errors for $Q_{tot}$ and $Q^*$, the PAC system is trained using three more loss functions: counterfactual assistance loss $L_{CA}$ for optimized value function factorization, information bottleneck loss $L_{LB}$ for succinct and effective assistive information generation, and local policy loss $L_{LP}$ for training factorized agent policy with entropy maximization.

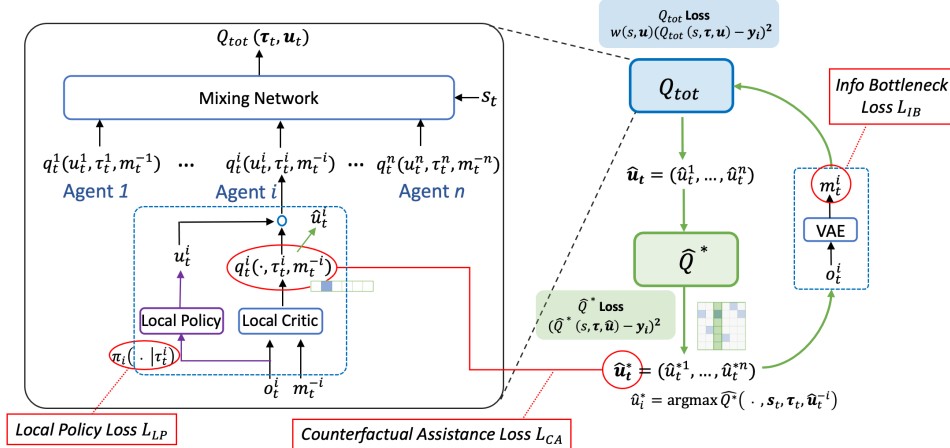

Figure 2: The overall architecture of PAC. With the help of assistive information, $m_t^{-i}$, counterfactual predictions $\hat{u}_t^*$ – which are obtained through an approximation of optimal $Q^*$ and an efficient local search – are used to directly train the per-agent value functions $q_t^i(\cdot, \tau_t^i, m_t^{-i})$ with respect to a new counterfactual assistance loss. It results in significantly improved value function factorization for partially-observable MARL problems.

**Generating Counterfactual Predictions** One of the key insights underlying this method is that the optimal joint action selection $u^*$ from the centralized $\hat{Q}^*$ can be used as a direct information assisting $Q_{tot}$ explicitly. Although the complexity of computing $\mathbf{u}^* = \arg\max_u \mathbb{E}[Q^*(s, \tau, \cdot)]$ is impractical as it grows in $O(|U|^n)$, however, it is possible to compute a local estimation of $\hat{\mathbf{u}}^*$ as $\hat{u}_i^* = \arg\max \hat{Q}^*(s, \boldsymbol{\tau}, \boldsymbol{u}_{-i}, \cdot)$, which reduces the computational complexity from exponential to linear level of $O(n \cdot |U|)$.

However, even with $\mathbf{u}^*$ provided, in what manner it can benefit value function factorization is not researched. Directly feeding this to the policy or critic network is counterproductive as the neural network can easily learn that this specific input is the key to reducing the TD-error while it is not helpful to explore or train the local policy and the training process might stall. Using an extra loss function, e.g. cross-entropy, as an effort to train the network to make decisions $u$ that are close to $\mathbf{u}^*$

sounds promising, however, it only quantifies the error without actually guiding the training since it may trap the policy in local sub-optimum and further limits the exploration process.

Inspired by difference rewards [16] and counterfactual baseline [17] for policy gradients, we propose a counterfactual assistance loss. For each agent $a$ we can use an advantage function that compares the $\hat{u}^*$ from $Q^*$ and $u$ from $Q^{tot}$ to a counterfactual baseline that marginalizes out the agent's potential optimal action which relegates $\hat{u}_a^*$, while keeping all other agents' action $\boldsymbol{u}^{-a}$ fixed, a counterfactual assistance loss is proposed as:

$$\mathcal{L}_{CA}(\mathbf{u}, \pi) = \sum_a \log\pi(u_i^a|\tau_i^a) \sum_{u_i \in [\boldsymbol{u}-u_i^*]} [q(\hat{u}_i^*, o_i, m_i) - \pi(u_i, o_i)q(u_i, o_i, m_i)] \tag{1}$$

This looks similar to calculating the counterfactual advantage as used in COMA, however, in COMA such counterfactual advantage is calculated for centralized critic which is computationally expensive and unstable which limits its performance, while our counterfactual assistance loss is providing direct guidance in training the policy network towards the direction of $u_i^*$. We later show in ablation studies that this loss is contributing to the performance improvements.

**Generating Assistive Information** Additionally, we show $\hat{\boldsymbol{u}}^*$ can be encoded and then used as input for local critics $q_i(u_i, \tau_i, m_i)$, as assistive information aiding value factorization. As previous works suggests [11, 18], we consider the information bottleneck method [19], with the Markov chain **o-m-$\hat{\mathbf{u}}^*$** during encoding. To be specific, we regard the internal representation of the intermediate layer as stochastic encoding $m_i$ of input source $o_i$, with the goal to learn an encoding that is maximally informative about the result $\hat{u}_i^*$. Formally, the objective for each agent $i$ can be written as:

$$J_{IB}(\boldsymbol{\theta}_m) = \sum_{j=1}^n \left[ I_{\theta_m}\left(\hat{u}_j^*; m_i|o_j, m_{-j}\right) - \beta I_{\boldsymbol{\theta}_m}(m_i; o_i) \right] \tag{2}$$

$m_i$ is an instance of random variable of assitive information that is drawn from multivariate Gaussian distribution $\boldsymbol{N}(f_m(o_i; \theta_m), \boldsymbol{I}))$, $\theta_m$ is the parameters of encoder $f_m$ and $\boldsymbol{I}$ is an identity matrix, a Lagrange multiplier parameter $\beta \geq 0$ is used to control the trade-off between the encoding the necessary information and reaching maximal compression. Yet this does not lead to a learnable model, since this mutual information $I$ is intractable. With the help of variational approximation, specifically, deep variational information bottleneck [20], we are able to parameterize this model using a neural network. We then derive and optimize a variational lower bound of such objective as

$$\mathcal{L}_{IB}(\boldsymbol{\theta}_m) = \mathbb{E}_{o_i \sim \mathcal{D}, m_j \sim f_m}[-\mathcal{H}[p(\hat{u}_j^*|\mathbf{o}), q_\psi(\hat{u}_j^*|o_j, \boldsymbol{m})] + \beta D_{\mathrm{KL}}(p(m_i|o_i)\|q_\phi(m_i))] \tag{3}$$

where $\mathcal{H}$ is the entropy operator, $D_{\mathrm{KL}}$ denotes Kullback-Leibler divergence operator and $q_\phi(m_i)$ is a variational posterior estimator of $p(m_i)$ with parameters $\phi$. Using the loss above a message encoder $f_m$ with parameters $\theta_m$ is trained to generate information $m_i \sim \boldsymbol{N}(f_m(o_i; \theta_m), \boldsymbol{I}))$ that is useful for decision making. Compared to [11, 18] that encodes the general state information and other agents' action selections as communication messages, which can not reduce the uncertainties in action-value functions; using the encoded $\hat{u}^*$ as assistance information can provide an explicit direction toward better individual value estimation and thus a joint value factorization. Detailed derivations and proofs can be found in Appendix A.1.

**Factorized Policy Iteration with Entropy Maximization** We leverage factorized policies to maintain decentralized execution in PAC despite the use of assistive information for training. Recent works have shown that Boltzmann exploration policy iteration is guaranteed to improve the policy and converge to optimal with unlimited iterations and full policy evaluation[21], within MARL domain it can be defined as: $J(\pi) = \sum_t \mathbb{E}\left[r(\mathbf{s}_t, \mathbf{u}_t) + \alpha\mathcal{H}(\pi(\cdot|\mathbf{s}_t))\right]$ where $\alpha$ denotes the temperature parameter that is used to adjust the balance between maximizing the entropy for a better exploration and maximizing the expected reward. We present one possible method of expanding this to MARL problems, to achieve decentralized policies in PAC and to encourage efficient exploration.

Several recent works are proposed to expand actor-critic or soft-actor-critic in to factorization based MARL methods [13, 12, 14], they all follow a centralized critic with decentralized actors (CCDA) framework. In this work, we train the actors in a centralized but factorized way. Unlike [14] that reuses the local utility network for both actor or critic or [13] which consists of a soft V-network and a soft Q-network for local policy net, we use a separate network as policy networks and propose a

centralized but factorized soft policy iteration with factorized Maximum-Entropy trained as:

$$\mathcal{L}_{LP}(\pi) = \mathbb{E}_{\mathcal{D}}\left[\boldsymbol{\alpha}\log\boldsymbol{\pi}\left(\boldsymbol{u}_t|\boldsymbol{\tau}_t\right) - Q_{tot}^{\tau}\left(\boldsymbol{s_t}, \boldsymbol{\tau_t}, \boldsymbol{u_t}, \boldsymbol{m_t}\right)\right]$$
$$= -q^{\text{mix}}\left(\boldsymbol{s}_t, \mathbb{E}_{\pi^i}\left[q^i\left(\tau_t^i, u_t^i, m_t^i\right) - \alpha^i\log\pi^i\left(u_t^i|\tau_t^i\right)\right]\right) \tag{4}$$

where $q^{mix}$ is the monotonic value decomposition network with $u_i \sim \pi_i(o_i)$ and $\mathcal{D}$ is the replay buffer used to sample training data.

As noted in previous research, choosing the temperature term in soft-actor-critic is non-trivial as it can vary in an unpredictable way when the policy becomes better as the training continues [13]. Instead of finding each individual temperature using approximation functions, since we use a centralized but factored policy, we consider one global temperature that is automatically adjusted by considering it as a constrained optimization problem as a parameter that is trained independently with loss:

$$\mathcal{J}(\alpha) = \mathbb{E}_{u_t \sim \pi_t}[-\alpha^i\log\pi_i(u_t|s_t) - \boldsymbol{\alpha}\mathcal{H}_0], \tag{5}$$

where $\mathcal{H}_0$ is a fixed entropy term so the temperature term $\alpha$ is generally decreasing such that the degree of exploration is reduced as the training proceeds[21].

**Training Process** So far we have discussed the components of our method, to formulate the new scalable RL algorithm, we now explain the implementation and the centralized training process for deep RL under DEC-POMDP. In the decentralized execution phase, only the policy (agent) network (marked as local policy in Fig.1 ) is enabled so that full CTDE is maintained.

Despite that monotonicity constraints limitations to the expressive power of the mixing network, recent research [14, 22] demonstrated that the IGM principle [5] as equivalent to joint greed actions and individual greedy actions is crucial, as it greatly promotes the sample efficiency; meanwhile most designs with only one unrestricted joint action-value function show poor empirical performance[5, 23], and thus we follow the design of WQMIX and keep both $Q_{tot}$ network and $\hat{Q}^*$ network.

The $\hat{Q}^*$ architecture (Green part in Fig. 1) is used as the estimator for $Q^*$ from unrestricted functions, where its mixing network is a feed-forward network that takes its local utilities. While the $Q_{tot}$ module (blue part in Fig.1) looks similar to QMIX and many other factorized methods, there is a significant difference, for its local estimator is provided with assistive information and the local value feeding to the network $q(u, \tau_i, m_i)$ is selected based on local policy, rather than taking $\text{argmax}\, q_i(\cdot, \tau_i, m_i)$. Then $Q_{tot}$ and $Q^*$ are trained with the objective to reduce their respective loss as:

$$\mathcal{L}_{\hat{Q}^*}(\theta) = \sum_i (\hat{Q}^*(s, \boldsymbol{\tau}, \hat{\boldsymbol{u}}) - y_i)^2. \tag{6}$$

$$\mathcal{L}_{Q_{tot}}(\theta) = \sum_i w(s, \mathbf{u})(Q_{tot}(s, \boldsymbol{\tau}, \mathbf{u}, \boldsymbol{m}) - y_i)^2 \tag{7}$$

where $\hat{u}_i = \text{argmax}\, q_i(\cdot, \tau_i, m_i)$, $y_i = r + \gamma\hat{Q}^*(s', \boldsymbol{\tau}', \text{argmax}_{\hat{\mathbf{u}}'} Q_{tot}(\boldsymbol{\tau}', \hat{\mathbf{u}}', s'; \boldsymbol{\theta}))$ with $\boldsymbol{\theta}^-$ being the parameters of the target network that are periodically updated to stabilize the training. $u_i \sim \pi_i(o_i)$ and $w(s, \mathbf{u})$ is the weighting function[1] to ensure the weighted projection can recover the correct maximal joint action value function for any Q[6]. Note the action selection for loss function in our method is different from the original design of QMIX and WQMIX. Apart from the independently updated entropy term $\alpha$, all other components (including the message encoder) are trained in an end-to-end manner with the objective to minimize the weighted sum of all losses proposed above, including the counterfactual loss and information bottleneck loss as

$$\mathcal{L}(\theta) = \mathcal{L}_{LP} + \mathcal{L}_{CA} + \mathcal{L}_{IB} + \mathcal{L}_{\hat{Q}^*} + \mathcal{L}_{Q_{tot}} \tag{8}$$

Detailed derivations can be found in Appendix A.1

## 5 Experiments

In this section, we compare the results with several state-of-the-art MARL methods on Predator-Prey [24] and selected StarCraft Multi-Agent Challenge (SMAC) [8] scenarios as benchmarks. More details on implementation, experiment settings, and hyperparameters are included in Appendix A.3. Code is available at github.com/hanhanAnderson/PAC-MARL.

---

[1]In this work we follow the weighting function design of OW-QMIX in WQMIX [6], as w = 1 if $Q_{tot}(s, \boldsymbol{\tau}, \mathbf{u}, \boldsymbol{m}) - y_i$ <0, w = $\alpha$ otherwise.

## 5.1 Cooperative Predator-Prey

We first consider Predator-Prey [24], which is a partially-observable multi-agent environment that involves 8 agents in cooperation as predators to catch 8 built-in-AI controlled units as prey on a 10x10 grid. Only when two or more predator agents surround and capture the same prey at the same time, it is considered a successful capture. We consider two settings: a simpler task without failed capture punishment and a harder task with a punishment reward of -2 when a capture attempt fails. This is to test the baseline algorithms on relative over-generalization and monotonicity constraint limitations. More details about this environment can be found in Appendix A.3.2.

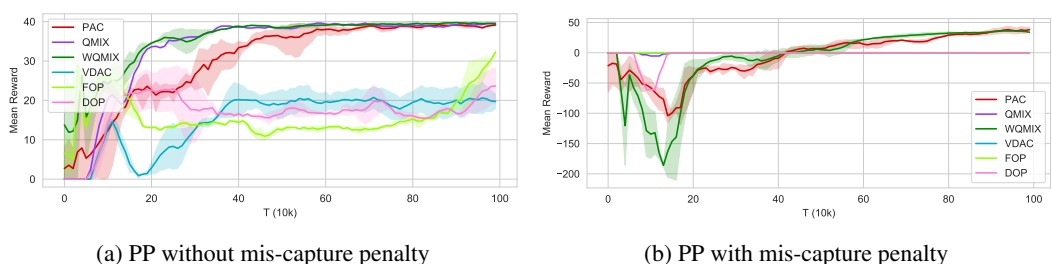

(a) PP without mis-capture penalty        (b) PP with mis-capture penalty

Figure 3: Results on Predator-Prey benchmark

As shown in Fig. 3a, for the easy task value-based method like QMIX, WQMIX can learn a policy that reaches the highest reward. However, in this environment, we see the performance gap between state-of-the-art value-based and policy-based methods since policy-based methods might suffer from relative over-generalization problem [9, 25, 26] or making a poor trade-off in joint policy representation. Our work as an actor-critic method can match the highest performance, although it takes a slightly longer time to converge, potentially due to the training of the variational inference models.

On the other hand, for the harder task that requires significant coordination among agents as shown in Fig. 3b, increased exploration and joint representation capabilities are required to finish the task which makes WQMIX and PAC the only two methods capable of learning a usable policy. Intuitively, when joint actions from uncoordinated decision-making occur more than coordinated ones, the penalty term will then dominate the average return from the environment and further the value estimation of each agent's local utility. We attribute these performance improvements to the use of entropy maximization which promotes more exploration attempts and the higher coordination abilities from assistive information.

## 5.2 SMAC

We then consider the SMAC [2] as the second benchmark, wherein each agent controls a unit cooperating with other friendly units in combat against the game's built-in AI-controlled units. The combat can be symmetric (same units for both parties) or asymmetric. Since it is shown that most state-of-the-art algorithms perform really well on easy and medium maps, which limits the demonstration of clear comparison and the potential improvements, we begin our test in six maps including two hard maps (5m_vs_6m, 3s_vs_5z) and four super-hard maps (MMM2, 27m_vs_30m, 6h_vs_8z, corridor). Selected maps are classified as hard or super hard due to (i) very large action space like 27m-vs-30m, (ii) requiring advanced exploration strategies like corridor (iii) requiring a higher level of coordination between the agents like 6h_vs_8z etc. We use the same default environment setting for all benchmark algorithms throughout the test. Each baseline algorithm is trained with 4 random seeds and evaluated every 10k training steps with 32 testing episodes. Details of the environment setup and hyper-parameter settings are listed in Appendix A.3.3. We choose state-of-the-art MARL algorithms as baseline including decomposed actor critic method: FOP [13] and DOP [12], decomposed policy gradient method: VDAC [14], decomposed value based method: QMIX [4], QPLEX [9]

---

[2]In this paper all SMAC experiments are carried out utilizing the latest SC2.4.10, performance is always not comparable across versions. We implemented our algorithm based on an open-sourced codebase [22] and acquired the results of QMIX and WQMIX from it.

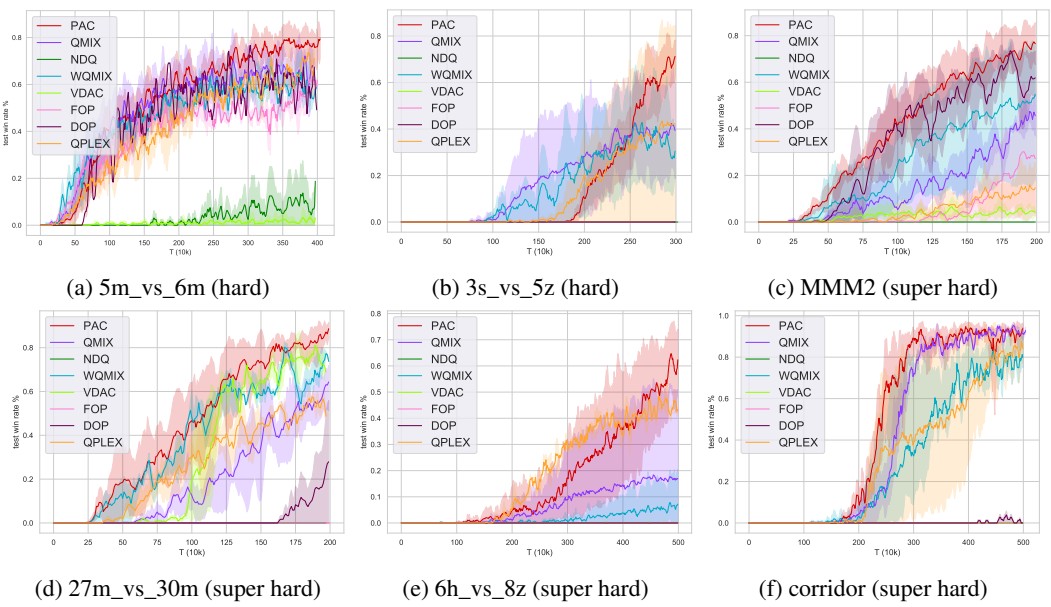

(a) 5m_vs_6m (hard)  (b) 3s_vs_5z (hard)  (c) MMM2 (super hard)

(d) 27m_vs_30m (super hard)  (e) 6h_vs_8z (super hard)  (f) corridor (super hard)

Figure 4: Results on SMAC benchmark

and WQMIX [6] ³ and a communication enabled value-based method: NDQ [11]. Results are shown in Fig.4. Overall, our method achieves the highest win rate compared to the baseline algorithms in terms of higher performance or faster convergence, especially on those maps that require more exploration and agents' coordination. As previous research demonstrated, there is a gap between SOTA value-based method and policy gradient methods, as the performance for most of them is limited on those maps that require extensive exploration techniques. Specifically, on `3s_vs_5z` and `6h_vs_8z`, PAC is able to train a usable policy that outperforms all baseline algorithms. On `corridor` PAC and the selected two value-based methods are able to learn a model, with our method converging faster with slightly better performance, while policy-based methods suffer from this map as it requires more exploration to find the specific trick in winning this challenging scenario. Finally on relatively easier maps, although most baseline algorithms hold a relatively close performance, the value-based and policy-based method performance gap still exists. Although FOP recently claimed to be the first multi-agent actor-critic method that outperforms state-of-the-art value-based methods on SMAC, we empirically found its limits when the chosen environment is substantially complicated and harder. We especially observe that our method as an actor-critic method has over-performed SOTA value-based MARL methods and brought significant improvements for actor-critic MARL.

## 5.3 Ablation Studies

We conduct ablation experiments to validate the effectiveness and contribution of each core component introduced in PAC on `MMM2` scenario in SMAC. Namely, in Fig.5 we consider verifying the effect of (1) optimization of temperature term in policy $\mathcal{J}(\alpha)$ by fixing $\alpha = 0.5$ as *PAC_fixedα*, (2) assisted information loss $\mathcal{L}_{IB}$ by disable it as *PAC_no_info*, (3) counterfactual assistance loss $\mathcal{L}_{CA}$ by replace it with a simple cross-entropy loss as *PAC_CE_ Loss*, (4) disable both $\mathcal{L}_{IB}$ and $\mathcal{L}_{CA}$ while substituting all $\hat{u}^*$ with $\hat{u}$ as *PAC_disabled* and (5) further remove the $Q^*$ structure from *PAC_disabled* as *PAC_No_Q\**. In this way, overall we can observe PAC outperforms ablated versions, especially by a large margin compared to *PAC_disabled* which indicates that a general encoding of the state infor-

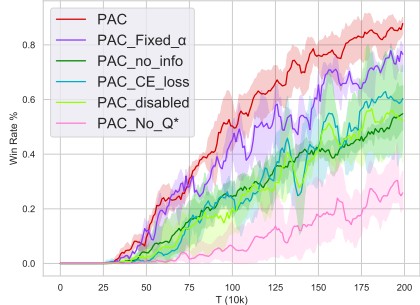

Figure 5: Ablations results comparing PAC and its hyperparameter-tuned ablated versions on SMAC map `MMM2`

---

³In this section we refer WQMIX to ow-qmix as it shows a general better performance than cw-qmix.

mation without explicit direction for training is not helpful
and may also harm the training.Although *PAC_CE_ Loss* later acquires competitive results, it takes a longer time, potentially due to the cross-entropy loss in this case trapped the training in a local sub-optimum. By fixing the entropy term $\alpha$ at 0.5 seems to be a balance point for maximizing the reward while promoting the exploration, yet it brings a performance drop, indicating the importance of updating the entropy term with its loss previously shown. *PAC_No_Q*$^*$ suffers from the most significant performance drop after most core components are removed from the original design. Such results validate how each component is crucial for achieving performance through experiments.

## 5.4 Limitations

Although the empirical results of PAC demonstrated improvements over both SOTA value-based and policy-based methods, at this stage, no strict convergence guarantee to the optimum is provided due to the scope of research. We also observe a higher variance in terms of the performance of PAC than value-based methods, this might be because of the entropy-maximization penalty. Also, we follow the tradition of parameter sharing among the agents, thus the role assignment of formulating distinctive behaving agents, which is another important topic was not considered.

## 6 Related Works

Cooperative multi-agent decision-making often suffers from exponential joint state and action spaces. Multiple approaches including independent Q-learning and mean-field games have been considered in the literature, while they do not perform well in challenging tasks or require homogeneous agents [14]. A paradigm of centralized training and decentralized execution (CTDE) has been proposed for scalable decision-making [27]. QPLEX [9] takes a duplex dueling network architecture to factorize the joint value function. Some of the key CTDE approaches include value function decomposition and multi-agent policy gradient methods.

Policy Gradient methods are considered to have more stable convergence compared to value-based methods [28, 29, 13] and can be extended to continuous action problems easily. A representative multi-agent policy gradient method is COMA [17], which utilizes a centralized critic module for estimating the counterfactual advantage of an individual agent. DOP [12] uses factorized policy gradients with architecture similar to Qatten[30]. However, as pointed out in [31, 23], multi-agent policy-based methods like MADDPG[32] are still outperformed by value-based methods StarCraft multi-agent challenge (SMAC) [8].

Decomposed actor-critic methods, which combine value function decomposition and policy gradient methods with the use of decomposed critics rather than centralized critics, are introduced to guide policy gradients. VDAC [14] combined the structure of actor-critic and QMIX for the joint state-value function estimation, while DOP [33] directly uses a network similar to Qatten [30] for policy gradients with off-policy tree backup and on-policy TD. The authors of [33] pointed out that decomposed critics are limited by restricted expressive capability and thus cannot guarantee the convergence of global optima; even though the individual policies may converge to local optima [13]. Extensions of the monotonic mixing function have also been considered, e.g., QTRAN [5] and weighted QMIX [6]. But solving tasks that require significant coordination remains a key challenge.

Another related topic is representational learning in reinforcement learning. A VAE-based forward model is proposed in [34] to learn the state representations in the environment. [35] considers a model to learn Gaussian embedding representations of different tasks during meta-testing. The authors in [36] proposed a recurrent VAE model which encodes the observation and action history and learns a variational distribution of the task. NDQ [11] encodes the state information as communication messages between agents. [37] use an inference model to represent the decision-making of the opponents. RODE [38] uses an action encoder to learn the action representations in restricting the role action spaces for a reduced policy search space. MAR [39] learns the metarepresentation for generalization problems. Unlike previous work, our method focuses on learning information from counterfactual predictions that are explicitly assistive for local estimation and efficient factorization.

## 7 Conclusions

In this paper, we propose PAC, a multi-agent framework utilizing extra state information as assistance for a better value function factorization, under a centralized but factored soft-actor critic setting. With the newly proposed counterfactual optimal joint action selection used for training and encoding assistance information, empirically we show our method not only matches or outperforms both state-of-the-art policy-based and value-based MARL algorithms on selected benchmarks but also bridges the gap between the two. Future work will also explore more effective ways to formulate and utilize extra state information to accelerate the training and tackle tasks in more complicated environments.

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
