# A  Appendix

## A.1  Mathematical Details

### A.1.1  Boundaries for counterfactual-prediction information

Following the introduction of the variational information bottleneck, we explain it as follows. Consider a Markov chain of $o - \hat{u*} - m$, (substituting the $X - Y - Z$ in the original IB and VIB), regarding the hidden representation of encoding $\hat{u*}$ of the input o, the goal of learning an encoding is to maximize the information about target m measured by the mutual information between encoding and the target $I(\hat{u*}; m)$.

To prevent the encoding of data from being $m = \hat{u*}$, which is not a useful representation, a constraint on the complexity can be applied to the mutual information as $I(\hat{u*}; m) \leq I_c$, where $I_c$ is the information constraint. This is equivalent to using the Lagrange multiplier $\beta$ to maximize the objective function $I(\hat{u*}; m) - \beta I(\hat{u*}; o)$. Intuitively, as the first term is to encourage $m$ to be predictive of $\hat{u*}$ while the second term is to encourage $m$ to forget $o$. Essentially $m$ is to act like a minimal sufficient statistic of $o$ for predicting $\hat{u*}$ [19].

Then specifically for each agent $i$, we intend to encourage assistive information$m_{-j}$ from other agents to agent $j$ to memorize its $\hat{u_j*}$ when assistive information from agent $i$ is conditioned on observation $o_j$, while we encourage assistive information $m_i$ from agent $i$ to not depend directly on its own observation $o_i$.

To efficiently and effectively encode extra state information for individual value estimation, we consider this information encoding problem as an information bottleneck problem [19], the objective for each agent $i$ can be written as:

$$J_{IB}\left(\boldsymbol{\theta}_m\right) = \sum_{j=1}^{n} \left[ I_{\theta_m}\left(\hat{u}_j^*; m_i | o_j, m_{-j}\right) - \beta I_{\boldsymbol{\theta}_m}\left(m_i; o_i\right) \right] \tag{9}$$

This object is appealing because it defines what is a good representation in terms of trade-off between a succinct representation and inferencing ability. The main shortcoming is that the computation of the mutual information is computationally challenging. Inspired by the recent advancement in Bayesian inference and variational auto-encoder [40, 37, 11], we propose a novel way of representing it by utilizing latent vectors from variational inference models using information theoretical regularization method, and then derive the evidence lower bound (ELBO) of its objective.

**Lemma 1.** *A lower bound of mutual information $I_{\theta_m}\left(\hat{u}_j^*; m_i | o_j, m_{-j}\right)$ is*

$$\mathbb{E}_{o_i \sim \mathcal{D}, m_j \sim f_m}[-\mathcal{H}(p(\hat{u}_j^*|\mathbf{o}), q_\psi(\hat{u}_j^*|o_j, \boldsymbol{m}))]$$

*where $q_\psi$ is a variational Gaussian distribution with parameters $\psi$ to approximate the unknown posterior $p(\hat{u}_j^*|o_j, m_j)$, $\mathbf{o} = \{o_1, o_2, \cdots, o_n\}$, $\mathbf{m} = \{m_1, m_2, \cdots, m_n\}$.*

*Proof.*

$$I_{\theta_m}\left(\hat{u}_j^*; m_i | o_j, m_{-j}\right)$$

$$= \int d\hat{u}_j^* do_j dm_{-j} p\left(\hat{u}_j^*, o_j, m_{-j}\right) \log \frac{p\left(\hat{u}_j^*, m_i | o_j, m_{-j}\right)}{p\left(\hat{u}_j^*|o_j, m_{-j}\right) p\left(m_i | o_j, m_{-j}\right)}$$

$$= \int d\hat{u}_j^* do_j dm_{-j} p\left(\hat{u}_j^*, o_j, m_{-j}\right) \log \frac{p\left(\hat{u}_j^*|o_j, m_{-j}\right)}{p\left(\hat{u}_j^*|o_j, m_{-j}\right)}$$

where $p(\hat{u}_j^*|o_j, m_{-j})$ is fully defined by our encoder and Markov Chain. Since this is intractable in our case, let $q_\psi(\hat{u}_j^*|\tau_j, m_{-j})$ be a variational approximation to $p(\hat{u}_j^*|o_j, m_{-j})$, where this is our decoder which we will take to another neural network with its own set of parameters $\psi$. Using the fact that Kullback Leibler divergence is always positive, we have

$$KL[p(\hat{u}_j^*|o_j, m_{-j}), q_\psi(\hat{u}_j^*|\tau_j, m_{-j})] \geq 0$$

$$\int d\hat{u}_j^* do_j dm_{-j} p(\hat{u}_j^*, o_j, m_{-j}) \log p(\hat{u}_j^*|o_j, m_{-j}) \geq \int d\hat{u}_j^* do_j dm_{-j} p(\hat{u}_j^*, o_j, m_{-j}) \log q_\psi(\hat{u}_j^*|o_j, m_{-j})$$

and hence

$$I_{\theta_c}\left(\hat{u}_j^*; m_i | o_j, m_j\right)$$

$$\geq \int d\hat{u}_j^* do_j dm_j p\left(\hat{u}_j^*, o_j, m_j\right) \log \frac{q_\psi\left(\hat{u}_j^* | o_j, m_j\right)}{p\left(\hat{u}_j^* | o_j, m_{-j}\right)}$$

$$= \int d\hat{u}_j^* do_j dm_j p\left(\hat{u}_j^*, o_j, m_j\right) \log q_\psi\left(\hat{u}_j^* | o_j, m_j\right) - \int d\hat{u}_j^* do_j dm_j p\left(\hat{u}_j^*, o_j, m_j\right) \log p\left(\hat{u}_j^* | o_j, m_{-j}\right)$$

$$= \int d\hat{u}_j^* do_j dm_j p(o_j) p(m_j|o_j) p(\hat{u}_j^*|o_j) \log q_\psi(\hat{u}_j^*|o_j, m_j) + \mathcal{H}(\hat{u}_j^*|o_j, m_j)$$

$$= \mathbb{E}_{\mathbf{o} \sim \mathcal{D}, m_j \sim f_m}\left(\int d\hat{u}_j^* p(\hat{u}_j^*|\mathbf{o}) \log q_\psi(\hat{u}_j^*|o_j, m_j)\right) + \mathcal{H}(\hat{u}_j^*|o_j, m_j)$$

$$= \mathbb{E}_{\mathbf{o} \sim \mathcal{D}, m_j \sim f_m}[-\mathcal{H}[p(\hat{u}_j^*|\mathbf{o}), q_\psi(\hat{u}_j^*|o_j, \boldsymbol{m})]] + \mathcal{H}(\hat{u}_j^*|o_j, m_j)$$

Notice that the entropy of labels $\mathcal{H}(\hat{u}_j^*|o_j, m_j)$ is an positive term that is independent of our optimization procedure and thus can be ignored. Then we have

$$I_{\theta_m}(\hat{u}_j^*; m_i | o_j, m_j) \geq \mathbb{E}_{\mathbf{o} \sim \mathcal{D}, m_j \sim f_m}[-\mathcal{H}[p(\hat{u}_j^*|\mathbf{o}), q_\psi(\hat{u}_j^*|o_j, \boldsymbol{m})]]$$

which is the lower bound of the first term in Eq.(2) $\qquad\square$

**Lemma 2.** *A lower bound of mutual information $I_{\boldsymbol{\theta}_m}(m_i; o_i)$ is*

$$\mathbb{E}_{T_i \sim D, m_j \sim f_m}[\beta D_{\mathrm{KL}}(p(m_i|o_i) \| q_\phi(m_i))]$$

*where $D_{\mathrm{KL}}$ denotes Kullback-Leibler divergence operator and $q_\phi(m_i)$ is a variational posterior estimator of $p(m_i)$ with parameters $\phi$.*

*Proof.*

$$I_{\boldsymbol{\theta}_m}(M_i; T_i)$$

$$= \int dm_i do_i p(m_i|o_i) p(o_i) \log \frac{p(m_i|o_i)}{p(m_i)}$$

$$= \int dm_i do_i p(m_i|o_i) p(o_i) \log p(m_i|o_i) - \int dm_i do_i p(m_i|o_i) p(o_i) \log p(m_i)$$

Again, $p(m_i)$ is fully defined by our encoder and Markov Chain, and when it is fully defined, computing the marginal distribution $\int do_i p(m_i|o_i) P(o_i)$ might be difficult. So we use $q_\phi(m_i)$ as a variational approximation to this marginal. Since $KL[p(m_i), q_\phi(m_i)] \geq 0$,

We have

$$\int dm_i p(m_i) \log p(m_i) \geq \int dm_i p(m_i) \log q_\phi(m_i)$$

Then

$$I_{\boldsymbol{\theta}_m}(M_i; T_i)$$

$$\leq \int dm_i do_i p(m_i|o_i) p(o_i) \log p(m_i|o_i) - \int dm_i do_i p(m_i|o_i) p(o_i) \log q_\phi(m_i)$$

$$= \int dm_i do_i p(m_i|o_i) p(o_i) \log \frac{p(m_i|o_i)}{q_\phi(m_i)}$$

$$= \mathbb{E}_{o_i \sim D, m_j \sim f_m}[D_{\mathrm{KL}}(p(m_i|o_i) \| q_\phi(m_i))]$$

$\qquad\square$

Combining **Lemma 1** and **Lemma 2**, we have the ELBO for the message encoding objective, which is to minimize

$$\mathcal{L}_{IB}(\boldsymbol{\theta}_m) = \mathbb{E}_{o_i \sim \mathcal{D}, m_j \sim f_m}[-\mathcal{H}[p(\hat{u}_j^*|\mathbf{o}), q_\psi(\hat{u}_j^*|o_j, \boldsymbol{m})] + \beta D_{\mathrm{KL}}(p(m_i|o_i) \| q_\phi(m_i))] \qquad (10)$$

### A.1.2 Factorized soft policy iteration

Recent works have shown that Boltzmann exploration policy iteration is guaranteed to improve the policy and converge to optimal with unlimited iterations and full policy evaluation, within MARL domain it can be defined as:

$$J(\pi) = \sum_t \mathbb{E}\left[r\left(\mathbf{s}_t, \mathbf{u}_t\right) + \alpha \mathcal{H}\left(\pi\left(\cdot|\mathbf{s}_t\right)\right)\right]$$

In section 4 we give the factorized soft policy gradients as:

$$\begin{aligned}
\mathcal{L}_{LP}(\pi) &= \mathbb{E}_{\mathcal{D}}\left[\alpha \log \boldsymbol{\pi}\left(\boldsymbol{u}_t|\boldsymbol{\tau}_t\right) - Q_{tot}^{\pi}\left(\boldsymbol{s_t}, \boldsymbol{\tau_t}, \boldsymbol{u_t}, \boldsymbol{m}_t\right)\right] \\
&= -q^{\mathrm{mix}}\left(\boldsymbol{s}_t, \mathbb{E}_{\pi^i}\left[q^i\left(\tau_t^i, u_t^i, m_t^i\right) - \alpha \log \pi^i\left(u_t^i|\tau_t^i\right)\right]\right)
\end{aligned} \tag{11}$$

We now derive it in detail, we use the aristocrat utility to perform credit assignment:

Let $q^{\mathrm{mix}}$ be the operator of a one-layer mixing network with no activation functions in the end whose parameters are generated from the hyper-network with input $\boldsymbol{s}_t$, then

$$q^{\mathrm{mix}}(\boldsymbol{s}_t, \boldsymbol{q}(\tau_t, a_t, m_t) - \boldsymbol{\alpha} \log \boldsymbol{\pi}(\boldsymbol{a_t}|\boldsymbol{\tau_t}))$$

$$= \sum_i [k^i(\boldsymbol{s})\mathbb{E}_{\pi}[q^i(\tau_t^i, a_t^i, m_t^i)] - \sum_i [k^i(\boldsymbol{s})\alpha^i \log \pi^i(a_t|\tau_t)] + b(\boldsymbol{s})$$

$\quad$ ( $k^i(s)$ and $b^i(s)$ are the corresponding weights and biases of $q^{\mathrm{mix}}$ conditioned on $\boldsymbol{s}$)

$$= \mathbb{E}_{\pi}[Q_{tot}(\boldsymbol{\tau}, \boldsymbol{a}, \boldsymbol{m}; \boldsymbol{\theta})] - \sum_i [k^i(\boldsymbol{s})\mathbb{E}_{\pi}[\alpha^i \log \pi^i(a_t|\tau_t)]$$

$$(q^{\mathrm{mix}}(\boldsymbol{s}_t, \boldsymbol{q}(\tau_t, a_t, m_t)) = \sum_i [k^i(\boldsymbol{s})\mathbb{E}_{\pi}[q^i(\tau_t^i, a_t^i, m_t^i)] + b(\boldsymbol{s}) = \mathbb{E}_{\pi}[Q_{tot}(\boldsymbol{\tau}, \boldsymbol{a}, \boldsymbol{m}; \boldsymbol{\theta})]\,)$$

$$(\mathbb{E}_{\pi}[Q_{tot}(\boldsymbol{\tau}, \boldsymbol{a}, \boldsymbol{m}; \boldsymbol{\theta})] = \sum_{\boldsymbol{a}} \pi^i(a_t^i|\tau_t^i)\mathbb{E}_{\pi}[Q_{tot}(\boldsymbol{\tau}, \boldsymbol{a}, \boldsymbol{m}; \boldsymbol{\theta})]\,)$$

$$= \mathbb{E}_{\pi}[Q_{tot}(\boldsymbol{\tau}, \boldsymbol{a}, \boldsymbol{m}; \boldsymbol{\theta})] - \sum_i \mathbb{E}_{\pi}[\boldsymbol{\alpha} \log \pi^i(a_t^i|\tau_t^i)]$$

$$(\text{let } \alpha^i = \frac{\boldsymbol{\alpha}}{k^i(\boldsymbol{s})})$$

$$= \mathbb{E}_{\pi}[Q_{tot}(\boldsymbol{\tau}, \boldsymbol{a}, \boldsymbol{m}; \boldsymbol{\theta})] - \sum_i \sum_{\pi} [\boldsymbol{\alpha}\pi^i(a_t^i|\tau_t^i) \log \pi^i(a_t^i|\tau_t^i)]$$

$$= \mathbb{E}_{\pi}[Q_{tot}(\boldsymbol{\tau}, \boldsymbol{a}, \boldsymbol{m}; \boldsymbol{\theta})] - \sum_{\pi} \boldsymbol{\alpha} \log \boldsymbol{\pi}(\boldsymbol{a}_t|\boldsymbol{\tau}_t)$$

$$(\text{Assume } \boldsymbol{\pi} = \prod \pi^i, \text{ then } \sum_i \sum_{\pi}[\boldsymbol{\alpha}\pi^i(a_t^i|\tau_t^i) \log \pi^i(a_t^i|\tau_t^i)] = \sum_{\pi} \boldsymbol{\alpha} \log \boldsymbol{\pi}(\boldsymbol{a}_t|\boldsymbol{\tau}_t))$$

$$= \mathbb{E}_{\pi}[Q_{tot}^{\pi}(\boldsymbol{s_t}, \boldsymbol{\tau_t}, \boldsymbol{a}) - \boldsymbol{\alpha} \log \boldsymbol{\pi}(\boldsymbol{a}_t|\boldsymbol{\tau}_t)]$$
$\qquad$ Which then complies to the original soft-actor-critic policy update policy.

We use the derivation above to show that directly using $\boldsymbol{q}(\tau_t, a_t, m_t) - \boldsymbol{\alpha} \log \boldsymbol{\pi}(\boldsymbol{a_t}|\boldsymbol{\tau_t})$ as input to feed in the mixing network to serve as soft-actor-critic policy update policy in a value decomposition method. It holds when using a single-layer mixing network without activation function, but nevertheless it offers insights of the proposed design, and when using relu activation function, it can be served as a lower bound object for optimization.

### A.2 Environment Details

We use more recent baselines (i.e., FOP and DOP) that are known to outperform QTRAN [5] and QPLEX [9] in the evaluation. In general, we tend to choose baselines that are more closely related to our work and most recent. This motivated the choice of QMIX (baseline for value-based factorization

---

**Algorithm 1** pseudocode for training PAC

---

1: **for** $k = 0$ to $max\_train\_steps$ **do**
2:     Initiate environment, critic network $q$, mixing network $Q^*, Q_{tot}$, policy network $\pi$, message encoder $m$
3:     Initiate Replay buffer $\mathcal{D}$
4:     **for** $t = 0$ to $max\_episode\_limits$ **do**
5:         For each agent $i$, take action $a_i \sim \pi_i$
6:         Execute joint action $\mathbf{a}$, observe reward $r$,
        and observation $\boldsymbol{\tau}$, next state $s_{t+1}$
7:         Store $(\boldsymbol{\tau}, \boldsymbol{a}, r, \boldsymbol{\tau}')$ in replay buffer $\mathcal{D}$
8:     **end for**
9:     **for** t = 1 to T **do**
10:         Sample trajectory minibatch $\mathcal{B}$ from $\mathcal{D}$
11:         Generate peer-assisted information
        $m_i \sim \boldsymbol{N}(f_m(o_i; \theta_m), \boldsymbol{I}))$, for $i = 0$ to $n$
12:         Calculate Loss
        $\mathcal{L}(\theta) = \mathcal{L}_{LP} + \mathcal{L}_{CA} + \mathcal{L}_{IB} + \mathcal{L}_{\hat{Q}^*} + \mathcal{L}_{Q_{tot}}$
13:         Update critic network and mixing network
        $\boldsymbol{\theta_{nn}}(\boldsymbol{q}, \boldsymbol{Q^*}, \boldsymbol{Q_{tot}}) \leftarrow \eta \hat{\nabla} \mathcal{L}(\theta)$
14:         Update policy network
        $\boldsymbol{\theta}(\pi) \leftarrow \eta \hat{\nabla} \mathcal{L}(\pi)$
15:         Update encoding network
        $\boldsymbol{\theta}_m(m) \leftarrow \eta \hat{\nabla} \mathcal{L}(\theta)$
16:         Update temperature parameter
        $\alpha \leftarrow \eta \hat{\nabla} \alpha$
17:         **if** $t$ mod $d = 0$ **then**
18:             Update target networks: $\boldsymbol{\theta}^- \leftarrow \boldsymbol{\theta}$
19:         **end if**
20:     **end for**
21: **end for**
22: Return $\boldsymbol{\pi}$

---

methods), WQMIX (close to our work that uses weighted projections so better joint actions can be emphasized), NDQ [11] (which similarly uses common information to assist decision making but as generating messages for agent-wise communication), VDAC [14], FOP [13], DOP [12] (SOTA actor-critic based methods). Our code implementation is available at Github.

### A.2.1 Multi-State Matrix Game

To highlight the importance of the extra state information for an assisted value function factorization, we present a multi-state matrix game as inspired by the single-state matrix game proposed in [5] and present how our method performs compared with the existing works.

The multi-state matrix game and the detailed mlearning results for more algorithms as shown in table 1 and table 2.

The multi-state matrix game can be considered the single state matrix game with the same goal of factorizing the global value, consider an Markov decision process (MDP) consisting of 2 states with 0.5 transition probabilities between them and two payoff matrices shown in table 1(a). Suppose that agent 1 has the same partial observation $o_1$ in states $s^{(1)}$ and $s^{(2)}$. Then, its per-agent value function $q_1(\cdot, \tau_1)$ computed from partial observation $o_1$ are also the same in both states. Due to the monotonicity of the mixing network (even though it is provided with complete joint state information), for any $u_1$ and $u_1'$ with ordering $q_1(u_1, \tau_1) \geq q_1(u_1', \tau_1)$ without loss of generality, we must simultaneously have $Q_{tot}(u_1, u_2, s^{(1)}) \geq Q_{tot}(u_1', u_2, s^{(1)})$ and $Q_{tot}(u_1, u_2, s^{(2)}) \geq Q_{tot}(u_1', u_2, s^{(2)})$ for any action $u_2$ of agent 2 in both states.

| x | **$a_2^1$** | $a_2^2$ | $a_1^1$ | | x | $a_2^1$ | $a_2^2$ | **$a_1^1$** |
|---|---|---|---|---|---|---|---|---|
| **$a_1^1$** | **4** | -2 | -2 | | $a_1^1$ | -2 | 0 | 0 |
| $a_1^2$ | -2 | 0 | 0 | | $a_1^2$ | **4** | -2 | -2 |
| $a_1^3$ | -2 | 0 | 0 | | **$a_1^3$** | -2 | 0 | 0 |

(a) Payoff matrix for state $s_1$ and $s_2$

| x | **0.3** | -1.2 | -2.6 | | x | **0.4** | -1.3 | -2.2 |
|---|---|---|---|---|---|---|---|---|
| -0.2 | 0.1 | -1.0 | -1.0 | | -0.2 | 0.4 | -1.0 | -1.0 |
| **0.3** | **1.1** | -0.9 | -1.0 | | **0.3** | **1.4** | -0.9 | -1.0 |
| -2.6 | -1.0 | -1.0 | -1.0 | | -2.6 | -1.0 | -1.0 | -1.0 |

(b) QMIX: $Q_{tot}(s_1)$, $Q_{tot}(s_2)$

| x | 1.0 | 0.2 | 0.2 | | x | 0.0 | 0.5 | **0.5** |
|---|---|---|---|---|---|---|---|---|
| 1.0 | **4.0** | -1.2 | -1.2 | | **1.0** | -1.2 | 0.1 | **0.1** |
| 0.0 | -0.2 | -1.6 | -1.6 | | 0.0 | -1.6 | -1.2 | -1.2 |
| 0.0 | -0.2 | -1.6 | -1.6 | | 0.0 | -1.6 | -1.2 | -1.2 |

(c) WQMIX: $Q_{tot}(s_1)$, $Q_{tot}(s_2)$

| x | 1.0 | 0.2 | 0.2 | | x | 0.0 | 0.5 | **0.5** |
|---|---|---|---|---|---|---|---|---|
| 1.0 | **4.0** | -1.8 | -1.9 | | **1.0** | -2.0 | 0.0 | **0.0** |
| 0.0 | -1.9 | 0.0 | 0.1 | | 0.0 | 0.1 | -2.0 | -2.0 |
| 0.0 | -2.1 | -0.1 | 0.0 | | 0.0 | -2.2 | -0.0 | -0.0 |

(d) WQMIX: $\hat{Q}^*(s_1)$, $\hat{Q}^*(s_2)$

| x | **0.7** | -2.0 | -2.1 | | x | **0.6** | -2.0 | -2.0 |
|---|---|---|---|---|---|---|---|---|
| **0.7** | **4.0** | -2.1 | -2.4 | | -1.2 | -1.8 | -2.5 | -2.5 |
| -2.0 | -2.1 | -2.4 | -2.4 | | **1.6** | **4.0** | -2.0 | -2.0 |
| -2.1 | -2.1 | -2.4 | -2.4 | | -1.8 | -2.1 | -2.5 | -2.5 |

(e) OURS: $Q_{tot}(s_1)$, $Q_{tot}(s_2)$

| x | **0.7** | -2.0 | -2.1 | | x | **0.7** | -2.0 | -2.1 |
|---|---|---|---|---|---|---|---|---|
| **0.7** | 4.0 | -2.1 | -2.1 | | -1.2 | -2.1 | -0.0 | -0.0 |
| -2.0 | -2.1 | -0.1 | -0.1 | | **1.6** | **4.0** | -2.0 | -2.0 |
| -2.1 | -2.1 | -0.1 | -0.1 | | -1.8 | -2.1 | -0.0 | -0.0 |

(f) OURS: $\hat{Q}^*(s_1)$, $\hat{Q}^*(s_2)$

Table 1: Payoff matrix of the one-step multi-state non-monotonic cooperative matrix game and reconstructed results from corresponding baselines. State $s_1$ and $s_2$ are selectet randomly on equal probability. Boldface indicates the local and joint optimal actions from local utilities and action-state value function.

| x | **0.7** | 0.2 | 0.2 | | x | **0.7** | 0.2 | 0.2 |
|---|---|---|---|---|---|---|---|---|
| 0.2 | 0.4 | 0.2 | -0.2 | | 0.2 | 0.4 | 0.2 | -0.2 |
| **0.7** | **0.7** | 0.4 | -0.4 | | **0.7** | **0.7** | 0.4 | -0.4 |
| 0.2 | 0.4 | 0.2 | 0.2 | | 0.2 | 0.4 | 0.2 | 0.2 |

(a) DOP: $Q_{tot}(s_1)$, $Q_{tot}(s_2)$

| x | **0.7** | 0.6 | -0.2 | | x | **0.7** | 0.6 | -0.2 |
|---|---|---|---|---|---|---|---|---|
| **1.1** | **1.7** | -0.9 | -0.1 | | **1.1** | **1.7** | -0.9 | -0.1 |
| 0.0 | -0.6 | 0.5 | -0.5 | | 0.0 | -0.6 | 0.5 | -0.5 |
| 0.0 | -0.6 | 0.5 | -0.5 | | 0.0 | -0.6 | 0.5 | -0.5 |

(b) FOP: $Q_{tot}(s_1)$, $Q_{tot}(s_2)$

| x | **0.7** | 0.2 | 0.2 | | x | **0.7** | 0.2 | 0.2 |
|---|---|---|---|---|---|---|---|---|
| **0.7** | **2.3** | 0.9 | 0.9 | | **0.7** | **3.2** | 2.2 | 2.2 |
| 0.2 | 2.3 | 0.9 | 0.9 | | 0.2 | 3.1 | 2.2 | 2.2 |
| 0.2 | 2.2 | 0.9 | 0.9 | | 0.2 | 3.1 | 2.2 | 2.2 |

(c) VDAC: $Q_{tot}(s_1)$, $Q_{tot}(s_2)$

| x | **0.5** | 0.3 | 0.2 | | x | **0.6** | 0.2 | 0.1 |
|---|---|---|---|---|---|---|---|---|
| 0.3 | 1.2 | -0.9 | -1.2 | | **0.5** | **0.7** | -1.2 | -0.0 |
| **1.2** | **1.2** | -1.0 | -0.5 | | 0.3 | 0.6 | -1.1 | -0.8 |
| 0.2 | -0.7 | -0.9 | -0.1 | | 0.2 | -2.1 | -0.1 | -1.0 |

(d) QTRAN: $Q_{tot}(s_1)$, $Q_{tot}(s_2)$

Table 2: Matrix results for other Benchmarks.

## A.2.2 Predator-Prey

A partially observable environment on a grid-world predator-prey task is used to model relative overgeneralization problem [24] where 8 agents have to catch 8 prey in a $10 \times 10$ grid. Each agent can either move in one of the 4 compass directions, remain still, or try to catch any adjacent prey. Impossible actions, i.e., moves into an occupied target position or catching when there is no adjacent prey, are treated as unavailable. If two adjacent agents execute the catch action, a prey is caught and both the prey and the catching agents are removed from the grid. An agent's observation is a 5 × 5 sub-grid centered around it, with one channel showing agents and another indicating prey. An episode ends if all agents have been removed or after 200 time steps. Capturing a prey is rewarded with r = 10, but unsuccessful attempts by single agents are punished by a negative reward p. In this paper we consider two sets of experiments with $p = 0$ and $p = -2$. The task is similart to matrix game proposed by [5] but significantly more complex, both in terms of the optimal policy and in the number of agents.

| map | Ally Units | Enemy Units |
|---|---|---|
| 1c3s5z | 1 Colossus, 3 Stalkers & 5 Zealots | 1 Colossus, 3 Stalkers & 5 Zealots |
| 3m | 3 Marines | 3 Marines |
| 3s5z | 3 Stalkers & 5 Zealots | 3 Stalkers & 5 Zealots |
| 8m | 8 Marines | 8 Marines |
| 3s_vs_5z | 3 Stalkers | 5 Zealots |
| 5m_vs_6m | 5 Marines | 6 Marines |
| MMM2 | 1 Medivac, 2 Marauders & 7 Marines | 1 Medivac, 3 Marauders & 8 Marines |
| 27m_vs_30m | 27 Marines | 30 Marines |
| 6h_vs_8z | 6 Hydralisks | 8 Zealots |
| corridor | 6 Zealots | 24 Zerglings |

Table 3: Brief Introduction of SMAC map scenarios used in experiments

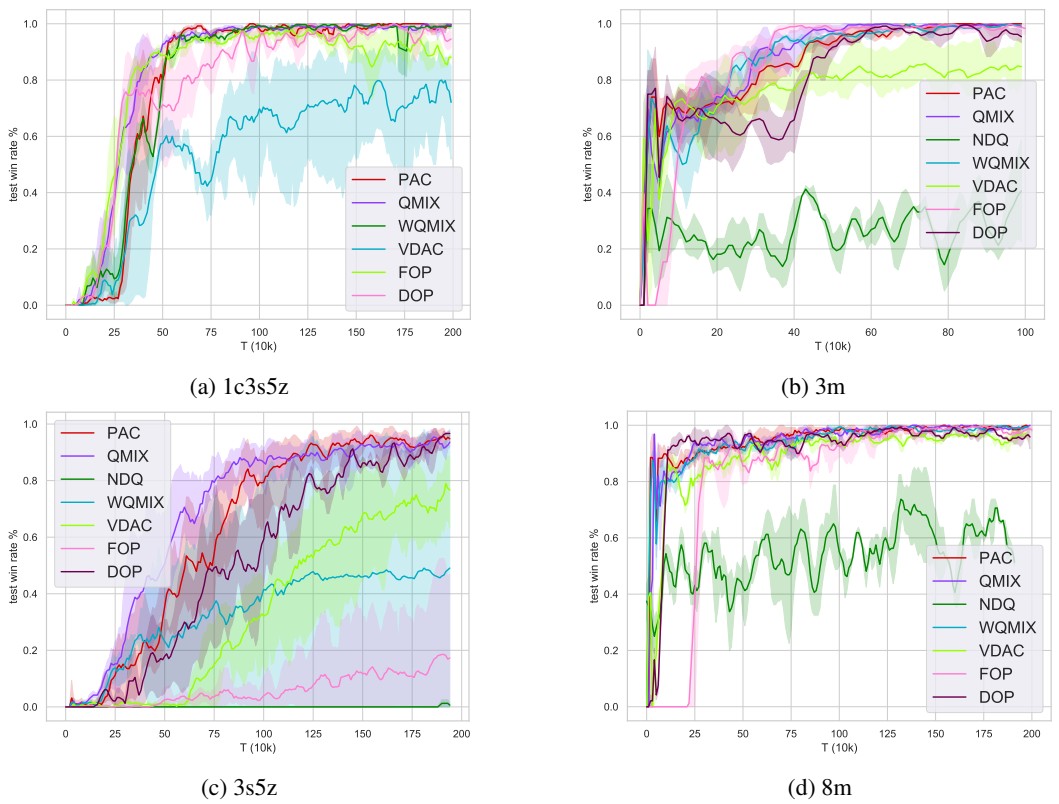

(a) 1c3s5z

(b) 3m

(c) 3s5z

(d) 8m

Figure 6: Additional results on SMAC benchmark

### A.2.3 SMAC

For the experiments on StarCraft II micromanagement, we follow the setup of SMAC [8] with open-source implementation including QMIX [4], WQMIX [6], NDQ [11], FOP [13], DOP [12] and VDAC [14]. We consider combat scenarios where the enemy units are controlled by the StarCraft II built-in AI and the friendly units are controlled by the algorithm-trained agent. The possible options for built-in AI difficulties are Very Easy, Easy, Medium, Hard, Very Hard, and Insane, ranging from 0 to 7. We carry out the experiments with ally units controlled by a learning agent while built-in AI controls the enemy units with difficulty = 7 (Insane). Depending on the specific scenarios(maps), the units of the enemy and friendly can be symmetric or asymmetric. At each time step each agent chooses one action from discrete action space, including noop, move[direction], attack[enemy_id] and stop. Dead units can only choose noop action. Killing an enemy unit will result in a reward

of 10 while winning by eliminating all enemy units will result in a reward of 200. The global state information are only available in the centralized critic. Each baseline algorithm is trained with 4 random seeds and evaluated every 10k training steps with 32 testing episodes for main results, and with 3 random seeds for ablation results and additional results. We carried out our experiment on a Nvidia GeForce RTX 2080 Ti workstation, on average it takes 3.5 hours to finish 3s5z map on SMAC environment for one run.

**Additional Results** We present additional results on easier maps including `1c3s5z`, `3m`, `3s5z`, and `8m` in Fig. 1.

|  | self-updating alpha | Q * Network | L_ca | L_ib |
|---|---|---|---|---|
| PAC | Y | Y | Y | Y |
| PAC_no_info | Y | Y | Y | |
| PAC_fixed_alpha | alpha = 1.0 | Y | Y | Y |
| PAC_CE_Loss | Y | Y | replace with CE | Y |
| **PAC_disabled*** | Y | Y | | |
| PAC_No_Q | Y | | | |

Table 4: Comparison of our method and its ablated versions

### A.2.4 Implementation details and Hyper-parameters

In this section we introduce the implementation details and hyper-parameters we used in the experiment. Recently [22] demonstrated that MARL algorithms are significantly influenced by code-level optimization and other tricks, e.g. using TD-lambda, Adam optimizer and grid-searched hyper-parameters (where many state-of-the-art are already adopted), and proposed fine-tuned QMIX and WQMIX, which is demonstrated with significant improvements from their original implementation. We implemented our algorithm based on its open-sourced codebase and acquired the results of QMIX and WQMIX from it. We use one set of hyper-parameters for each environment, i.e., no tuned hyper-para for individual maps. Unless otherwise mentioned, we keep the same setting for common hyper-parameters shared by all algorithms, e.g. learning rate, and keep their unique hyper-parameters to their default settings.

We use epsilon greedy for action selection with annealing from $\epsilon = 0.995$ decreasing to $\epsilon = 0.05$ in 100000 training steps in a linear way.

Batch size $bs = 128$, replay buffer size = 10000

Target network update interval: every 200 episodes

$\beta = 0.001$, since $o_i$ and $\hat{u}_i^*$ are within similar dimensions and thus does not require very high compression.

Weights $w = 0.5$ in weighting functions.

learning rate $lr = 0.001$

td lambda $\lambda = 0.6$

initial entropy term $\log\alpha = -0.07$, with its learning rate $lr_\alpha = 0.0003$

performance for each algorithm is evaluated for 32 episodes every 1000 training steps.

### A.2.5 Hyperparameter-tuning for ablation studies

To fully demonstrate the effectiveness of each components, we performed hyperparameter-tuning for each ablated settings. Specifically, we choose exploration steps (epsilon anneal time = [50k, 100k], where $\epsilon$ decays from 0.095 to 0.05 in $\epsilon$- greedy), eligibility traces ($\lambda = [0.3, 0.5, 0.6]$ in TD-$\lambda$), and replay-buffer size (buffer = [5000, 10000]) and use the results with best performance to serve as the ablated results. We show the optimal hyperparameter setting for each ablated versions and their corresponding final winning rates on SMAC environment as in Table 4.

| Setting | Optimal Hyperparameter Setting | Average Winning Rates |
|---|---|---|
| PAC | $\lambda$=0.5 buffer_size=10000, episilon_anneal_time=100000 | 0.87 |
| PAC_Fixed_alpha | $\lambda$=0.6 buffer_size=5000, episilon_anneal_time=80000 | 0.80 |
| PAC_no_info | $\lambda$=0.6 buffer_size=10000, episilon_anneal_time=100000 | 0.51 |
| PAC_CE_loss | $\lambda$=0.5 buffer_size=10000, episilon_anneal_time=100000 | 0.61 |
| PAC_disabled | $\lambda$=0.5 buffer_size=5000, episilon_anneal_time=80000 | 0.49 |
| PAC_No_Q* | $\lambda$=0.6 buffer_size=10000, episilon_anneal_time=100000 | 0.25 |

Table 5: Optimal Hyperparameter Setting for ablated versions