# OpenReview forum: "PAC: Assisted Value Factorization with Counterfactual Predictions in Multi-Agent Reinforcement Learning"
_NeurIPS.cc/2022/Conference — NeurIPS 2022 Accept_

### Official Review · Reviewer_QNBi · 2022-07-01

**Rating:** 4
**Confidence:** 3
**Soundness:** 2 fair
**Presentation:** 2 fair
**Contribution:** 2 fair

**Summary:**

This paper aims to improve the joint-action Q-values learnt in a cooperative MARL setting under the paradigm of CTDE. The authors aim to do so by first identifying the shortcomings of existing methods, QMIX and WQMIX in particular, under significant partial observability. To remedy the errors, during the centralised training they allow for agents to communicate and explicitly encourage them to send messages that improve the learning of the optimal actions for each agent.

**Questions:**

1. Could you expand on the derivation on Eqn 4 (more details on specific questions are in the weaknesses section).

2. It seems that NDQ performs terribly in your SMAC experiments, is that expected?

3. What weighting function are you using for the WQMIX baseline and your algorithm in Eqn 7?

### Minor:

Abstract - Monotonicity isn't required for maximisation (e.g. QPLEX).

Line 191 - The justification for using the maximum entropy framework is a little misleading. It can help exploration in practise, but to prove it converges the paper uses full policy evaluation and iteration which is generally not feasible in a practical setting.

Footnote 1 pg6 - I don't think the built-in AI is really considered harder than the prev version. The strength of the units has changed (colossus in particular), which makes some maps much easier.


**Limitations:**

-

**Strengths And Weaknesses:**

## Strs

Section 3 is good for motivating the central issue the paper aims to address. It could be made clearer that these are the joint-action $Q$-values learnt during the centralised training where communication *is* allowed for PAC (but not for the other methods).


## Weaknesses

No mention of QPLEX in this paper at all (except for a reference related to over-generalisaiton). At the very least it should be in the related works section.

Sending messages to be maximally informative about which action to take is a nice idea. However, to really drive this point home there should be an ablation where this part of the loss is turned off (apologies if there already is, it was unclear in Section 5.3).

I didn't follow the derivation of Eqn 4 in the Appendix. What is $k^i(s)$? Did you mean to have the expectation be over $\pi^i$ for the first line?
Are you missing a summation over all possible joint-actions in the 3rd line?
Why expand the expectation going from line 2 to 3, when you just bring it back in line 5?
How do you get to the last line from the penultimate one? The expectation changes from $\pi$ to $\pi^i$ and you also introduce $q^{mix}$ and drop the $b(s)$.

The results look fine, but I would have liked to have seen another environment that exhibited the same kind of issues as Fig 1.
From the results it appears that the issues due to partial observability aren't that much of a concern in these benchmarks.
Is that the expected outcome? What kind of settings would significantly benefit from your approach (in the sense that something like WQMIX fails completely but yours doesn't)?

The weighting function you are using for the WQMIX baseline and your algorithm in Eqn 7 should be described.

---

> ### Author Response · Authors · 2022-08-02
> **Response to Reviewer QNBi**
>
> We express our deepest appreciation for your time and insightful comments.
>
> > No mention of QPLEX in this paper at all (except for a reference related to over-generalization). At the very least it should be in the related works section.
>
> * We have updated the related work introducing the design and advances of different works including QPLEX, WQMIX and QTRAN. Regarding QPLEX in particular, a later work FOP [6] – which we have selected as a baseline in the experiment results –  shows a higher performance than QPLEX in almost all scenarios. Thus, In the evaluation, we have decided to use the more recent FOP as a baseline.
>
> > Sending messages to be maximally informative about which action to take is a nice idea. However, to really drive this point home there should be an ablation where this part of the loss is turned off (apologies if there already is, it was unclear in Section 5.3).
>
> * Thank you for the suggestion. As also pointed out by another reviewer, the current version of ablation studies can be confusing. We have redesigned the ablation studies, provided new results, and rewrote the ablation section. By comparing the original design and the assistive information-disabled design, the performance gap verifies its necessity. A full comparison of PAC and its ablated versions of our deisgn is as below. Hopefully, this would better illustrate the contribution of each component in our design.
>
>
> |                      | **self-updating alpha** | **Q * Network** | **L_ca**            | **L_ib** |
> |----------------------|-------------------------|----------------|---------------------|----------|
> | **PAC**              | **√**                   | **√**          | **√**               | **√**    |
> | **PAC_no_info**          | **√**                   | **√**          | **√**               |          |
> | **PAC_fixed_alpha**        | **alpha = 1.0**         | **√**          | **√**               | **√**    |
> | **PAC_CE_Loss**          | **√**                   | **√**          | **replace with CE** | **√**    |
> | **PAC_disabled* | **√**                   | **√**          |                     |          |
> | **PAC_No_Q***            | **√**                   |                |                     |          |
>
>
> > I didn't follow the derivation of Eqn 4 in the Appendix. What is $k^i(s)$? Did you mean to have the expectation be over $\pi^i$ for the first line? Are you missing a summation over all possible joint-actions in the 3rd line? Why expand the expectation going from line 2 to 3, when you just bring it back in line 5? How do you get to the last line from the penultimate one? The expectation changes from $\pi$ to  and you also introduce $\pi^i$ and drop the $b(s)$.
>
> * Realizing that the current derivation of eq4 can be hard to follow, we have updated the derivation of eq4 in the revised version of the appendix with assumptions used and between-step explanations. We use this derivation to show that directly using $\boldsymbol{q} \boldsymbol{(\tau_{t}, a_{t},m_{t})} - \boldsymbol{\alpha} \log \boldsymbol{\pi(a_{t}| \tau_{t}})$ as input to feed in the mixing network to serve as soft-actor-critic policy update policy in a value decomposition method.
>
> > The results look fine, but I would have liked to have seen another environment that exhibited the same kind of issues as Fig 1. From the results it appears that the issues due to partial observability aren't that much of a concern in these benchmarks. Is that the expected outcome? What kind of settings would significantly benefit from your approach (in the sense that something like WQMIX fails completely but yours doesn't)?
>
> * Response: In our evaluation, we have not seen a case in SMAC where WQMIX fails completely, but there are multiple cases where WQMIX and QMIX are suffering significant performance losses.  For instance, maps like 6h_vs_8z and 3s_vs_5z that require higher coordination between agents. Another scenario is when $u$ learned is at bad local optima that are away from $u^*$, e.g. due to relative overgeneralization, counterfactual assistance loss could help so it enjoys the benefits of sample efficiency from the monotonic mixing network while preventing the harm from its representational limitations. Note PAC is not a communication-based method and thus cannot provide real-time information to fundamentally resolve the partial observability issue, it provides additional information for better utilizing the state information for better value factorization.
>
> > The weighting function you are using for the WQMIX baseline and your algorithm in Eqn 7 should be described.
>
> * We have updated the weighting function in the revised version (marked in blue color). Specifically, it is w = 1 if TD-error <0, otherwise w = $\alpha$ as we are following the design of ow-qmix.

---

> > ### Author Response · Authors · 2022-08-02
> > **Response to Reviewer QNBi (cont'd )**
> >
> > > Reviewer: Q1 Could you expand on the derivation on Eqn 4 (more details on specific questions are in the weaknesses section).
> >
> > * Response: We have updated the derivation of eq4 (marked in color blue) in the revised version with assumptions used and between-step explanations in the appendix.
> >
> > > Reviewer: Q2 It seems that NDQ performs terribly in your SMAC experiments, is that expected?
> >
> > * Response: The main reason we pick NDQ as one of our baseline algorithms is that NDQ is another MARL work using mutual information but as agent-wise communication message generation. Its performance is somewhat as expected. First, NDQ has been shown to converge somewhat slower than baselines we choose, e.g. in Fig.7 of their paper for map 3s_vs_5z it starts to show non-zero test win rates after more than 5M training steps, while we only compare the results for the first 3M training steps since PAC and some other baselines are able to achieve a high win rate at 3M steps already. Second,  NDQ performs well in environments that need real-time agent-wise communication, e.g. on environments like hallway and  3b_vs_1h1m in SMAC, while it is absent in our testing benchmarks, as we consider mainly hard maps in our evaluation.
> >
> >
> >
> >
> > > Reviewer: What weighting function are you using for the WQMIX baseline and your algorithm in Eqn 7?
> >
> > * Response: We use the same weighting function as in ow-qmix from WQMIX, specifically, it is w = 1 if TD-error <0 otherwise w = $\alpha$. We have updated the weighting function in the revised version (marked in blue color).
> >
> >
> > > Reviewer: Minor: Abstract - Monotonicity isn't required for maximisation (e.g. QPLEX).
> >
> > * Response:  We were meant to introduce by saying "qmix-based value function factorization methods". We have now removed this in the revised version.
> >
> > >  Reviewer:  Minor:  Line 191 - The justification for using the maximum entropy framework is a little misleading. It can help exploration in practise, but to prove it converges the paper uses full policy evaluation and iteration which is generally not feasible in a practical setting.
> >
> > * Response:  We update the expression to “Recent works have shown that Boltzmann exploration policy iteration is theoretically guaranteed to improve the policy and converge to optimal with unlimited iterations and full policy evaluation”
> >
> > >  Reviewer:  Footnote 1 pg6 - I don't think the built-in AI is really considered harder than the prev version. The strength of the units has changed (colossus in particular), which makes some maps much easier.
> >
> > * Response:  We change the footnote to " Performance is always not comparable across versions." instead. After consulting with other authors and the SC2 change log, we find that some maps became easier and some became harder, as compared to the old version.
> >
> >
> >
> >
> >
> >
> >
> > [1] Hu, Jian, et al. "Rethinking the implementation tricks and monotonicity constraint in cooperative multi-agent reinforcement learning." arXiv e-prints (2021): arXiv-2102.

---

> > > ### Comment · Reviewer_QNBi · 2022-08-05
> > > **Derivation**
> > >
> > > It would be worth making explicitly clear in the main paper that your derivation assumes a linear mixing function (as opposed to a more general monotonic one), and that it only holds for specific choices of $\alpha$.
> > > It also looks like $\alpha$ should be a vector in the first line, and then you should have $\alpha_i$ for the second line of Eq 4.

---

> > > > ### Author Response · Authors · 2022-08-08
> > > > **Response to Reviewer Reply**
> > > >
> > > > Thank you for your response. We have revised the paper with suggested clarification, e.g., regarding $\alpha$. We would like to clarify that while the derivation is for a linear mixing function (which is now clearly described in derivation), it provides insights into our general design - adding entropy-regularization in local critics to serve as input to feed in the mixing network as soft-actor-critic policy update policy in a value decomposition method.

---

> > > > ### Author Response · Authors · 2022-08-09
> > > > **Response to Reviewer Reply**
> > > >
> > > > Dear reviewer, thank you again for your valuable time and positive suggestions that will make our paper stronger. In addition to the previous response, we have the results of QPLEX added and redesigned the ablation studies section so it is more clear for demonstration now. We also revised the paper with more clarification regarding the expression, notations, and explanations of the equation derivations. Please consider raising our score, or let us know if there are other places that need addressing. Thank you!

---

### Official Review · Reviewer_i5it · 2022-07-07

**Rating:** 5
**Confidence:** 3
**Soundness:** 3 good
**Presentation:** 2 fair
**Contribution:** 3 good

**Summary:**

This paper leverages Assistive information by Counterfactual Predictions of optimal joint action selection, which use a novel counterfactual loss to assist value function factorization. The paper develops a variational inference-based information encoding method to collect and encode the counterfactual predictions from an estimated baseline with the Markov chain o-m-ˆu∗.

**Questions:**

On page 3, line 106, the paper says that there are the same $$ in different states $ s $. Can you explain why?

Whether the conditional mutual information I(u_j;m_i|o_j,m_-j) given in formula (2) still conforms to the original theory.


**Strengths And Weaknesses:**

originality：
This paper uses a deep variational bottleneck to encode the counterfactual optimal joint action selection. our loss functions are set up to assist in the value decomposition between agents. This method is novel. The training assistive information is generated using the variational inference method to expand the representational ability of value functions.

quality, clarity:
The article is well written. Some symbols need to be clearly defined.

significance:
This paper introduces the information bottleneck into the multi-agent cooperative task, which is of significance to the application of the information bottleneck method in this field.

---

> ### Author Response · Authors · 2022-08-02
> **Response to Reviewer i5it**
>
> We thank the reviewer for their constructive and insightful comments and suggestions.
>
> > Reviewer: Question: On page 3, line 106, the paper says that there are the same $$ in different states s . Can you explain why?
>
> * Response: We assume the reviewer is asking why there are the same observations $o$ in different states $s$. Note that this is a new multi-state matrix game with partial observability. Agent 1 has only partial observations and thus cannot fully distinguish state 1 or state 2 from its local observations (i.e. $o_{1}(s_{1})$ = $o_{1}(s_{2})$). This is a simplified scenario of a more common case in POMDP where one agent may partially observe state transitions. We introduce such an example to illustrate the importance of the need for additional assistive information in POMDP. Existing methods cannot effectively factorize the pay-off matrix in this game.
>
> > Reviewer: Question: Whether the conditional mutual information I(u_j;m_i|o_j,m_-j) given in formula (2) still conforms to the original theory.
>
> * Response: Yes, it still conforms to the original theory. Following the introduction of the variational information bottleneck, we explain it as follows.
> Consider a Markov chain of  $o - \hat{u*}-m$,  (substituting the $X-Y-Z$ in the original IB and VIB), regarding the hidden representation of encoding $\hat{u*}$ of the input o, the goal of learning an encoding is to maximize the information about target m measured by the mutual information between encoding and the target $I(\hat{u*};m)$.
>
> To prevent the encoding of data from being $m = \hat{u*}$, which is not a useful representation, a constraint on the complexity can be applied to the mutual information as $I( \hat{u*}; m) \leq I_{c}$, where $I_{c}$ is the information constraint. This is equivalent to using the Lagrange multiplier $\beta$ to maximize the objective function $ I( \hat{u*}; m) - \beta  I( \hat{u*}; o)$. Intuitively, as the first term is to encourage $m$ to be predictive of $\hat{u*}$ while the second term is to encourage $m$ to forget $o$. Essentially $m$ is to act like a minimal sufficient statistic of $o$ for predicting $\hat{u*}$ [1].
>
> Then specifically for each agent $i$, we intend to encourage assistive information$m_{-j}$  from other agents to agent $j$ to memorize its $\hat{u_j*}$ when assistive information from agent $i$ is conditioned on observation $o_{j}$,
> while we encourage assistive information $m_i$ from agent $i$ to not depend directly on its own observation $o_i$. Then we have the definition of assistive information generation as eq(2). Following [2], we have a neural network to generate such assistive information with the objective as the evidence lower bound derived from it in appendix A1.1.
> Note in this specific task $o$ is not at a higher dimension of information compared to $\hat{u*}$, thus a small $\beta$ can be used as we described in the appendix when listing hyper-parameters used.
> We will add the explanations above to the appendix in the revised version for a coherent presentation.
>
>
>
> > Reviewer: quality, clarity: The article is well written. Some symbols need to be clearly defined.
>
> * Response: Thank you. We have proofread our paper and introduced additional explanations and definitions to the symbols used in the revised version.
>
> [1] Tishby, Naftali, Fernando C. Pereira, and William Bialek. "The information bottleneck method." arXiv preprint physics/0004057 (2000).
>
> [2] Alemi, Alexander A., et al. "Deep variational information bottleneck." arXiv preprint arXiv:1612.00410 (2016).

---

> ### Author Response · Authors · 2022-08-08
> **Response to Reviewer**
>
> Thank you for your review. We hope our response has solved your doubts.

---

> ### Author Response · Authors · 2022-08-09
> **Response to Reviewer**
>
> Thank you for your review. We hope our response has solved your doubts. If you think these revisions are helpful for making the paper a better one, please consider raising our score, or let us know if there are other places that need addressing. Thank you!

---

### Official Review · Reviewer_cYxS · 2022-07-11

**Rating:** 5
**Confidence:** 4
**Soundness:** 3 good
**Presentation:** 3 good
**Contribution:** 3 good

**Summary:**

The paper proposes that in partially observable MARL problems, the agent's ordering of its own action may impose concurrency constraints on the representation function, resulting in estimation error at training time. To address this problem, the authors propose that PAC uses counterfactuals selected from the best joint action. The auxiliary information generated in the prediction is explicitly aided by a new counterfactual loss on the decomposition of the value function. Variational inference-based information encoding is also proposed to collect and encode the counterfactual predictions from the estimated baseline. To achieve distributed decision making, thus deriving a factorization strategy for each agent inspired by the maximum entropy MARL framework, and the benefits of the algorithm are experimentally illustrated.

**Questions:**

* Is it possible to design further ablation experiments to illustrate the specific effects of the different parts? Instead of just performance, I am curious about the actual impact of the different designs in the experiments.
* Are the hyperparameters used in the ablation experiments the same parameters used in the training? Was fine tuning of the hyperparameters done after the ablation? Intuitively, if the original training hyperparameters were used, then removing a certain part would definitely affect the training performance, so further hyperparameter tuning would be needed.
* The experimental results in the paper do not look too good, does it come from the detailed differences in code implementation? The difference in code implementation has been widely concerned in the ma field, so I suggest the authors use a more fair codebase for comparison, e.g., https://github.com/hijkzzz/pymarl2

**Ethics Review Area:**

["I don’t know"]

**Strengths And Weaknesses:**

Strengths：
* paper writing is clear and better understood
* The motivation is clear and easy to understand
* The introduction of matrix game example is very intuitive
* Many comparison methods in the experimental part
* There are ablation experiments to illustrate the effect of different parts on performance

Weaknesses：
* The ablation section just tests the impact of different parts on the performance, without further analysis of what is the specific manifestation of the impact of each part?
* The authors claim that their method is SOTA, but the performance of the experiment is puzzling, for example, the performance on smac environment does not look good. Are there some differences in code implementation details.

---

> ### Author Response · Authors · 2022-08-02
> **Response to Reviewer cYxS**
>
> Thank you for the constructive comments. We will follow these helpful comments in our revised version. Following are our responses to your questions and concerns.
>
> > Reviewer: Is it possible to design further ablation experiments to illustrate the specific effects of the different parts? Instead of just performance, I am curious about the actual impact of the different designs in the experiments.
> Weaknesses：The ablation section just tests the impact of different parts on the performance, without further analysis of what is the specific manifestation of the impact of each part?
>
> * Response: We appreciate the comments/suggestions regarding the ablation section. We have redesigned our ablation experiments, added new results in the figure, and rewrite the section with a more clearer description and explanation of each test setting in the revised version. To verify the effect of (1) optimization of temperature term in policy $\mathcal{J}(\alpha)$ by fixing $\alpha= 0.5$ as \textit{PAC\_fixed}${\alpha}$, (2) assisted information loss $L_{IB}$ by disable it as {PAC\_no\_info},  (3) counterfactual assistance loss $L_{CA}$ by replace it with a simple cross-entropy loss as PAC\_CE\_ Loss, (4) disable both $L_{IB}$ and $\mathcal{L}_{CA}$ while substituting all $\hat{u}^*$ with $\hat{u}$ as PAC\_disabled and (5) further remove the $Q^*$ structure from PAC\_disabled as PAC\_No\_$Q^*$.  To be specific, a detailed comparison chart is as below:
>
> |                      | **self-updating alpha** | **Q * Network** | **L_ca**            | **L_ib** |
> |----------------------|-------------------------|----------------|---------------------|----------|
> | **PAC**              | **√**                   | **√**          | **√**               | **√**    |
> | **PAC_no_info**          | **√**                   | **√**          | **√**               |          |
> | **PAC_fixed_alpha**        | **alpha = 1.0**         | **√**          | **√**               | **√**    |
> | **PAC_CE_Loss**          | **√**                   | **√**          | **replace with CE** | **√**    |
> | **PAC_disabled* | **√**                   | **√**          |                     |          |
> | **PAC_No_Q***            | **√**                   |                |                     |          |
>
>
> > Reviewer: Are the hyperparameters used in the ablation experiments the same parameters used in the training? Was fine tuning of the hyperparameters done after the ablation? Intuitively, if the original training hyperparameters were used, then removing a certain part would definitely affect the training performance, so further hyperparameter tuning would be needed.
>
> * Response: As we noted in the paper and in the appendix, we keep the same set of hyperparameters for all scenarios and experiments of SMAC, including in both main results and ablation studies. As noted by the reviewer, differences in code implementation and hyperparameter tuning are widely concerned in the field. It is possible that conducting a hyper-parameter tuning could further improve the performance of our algorithm. Our logic is that since no specific hyper-parameter tuning is performed on any algorithms in our evaluation, it still provides a fair comparison between different algorithms. Thus, we can verify that the improvements are indeed coming from our new design rather than non-algorithm-related training tricks.
>
>
> > Reviewer: The experimental results in the paper do not look too good, does it come from the detailed differences in code implementation? The difference in code implementation has been widely concerned in the ma field, so I suggest the authors use a more fair codebase for comparison, e.g., https://github.com/hijkzzz/pymarl2
>
> * Response: In fact, in this paper, we implemented our algorithm based on pymarl2 (https://github.com/hijkzzz/pymarl2) exactly as suggested by the reviewer, and obtained evaluation results for QMIX, WQMIX from it. It is why QMIX and WQMIX baselines are performing better in our figures than reported in some previous work. Although some other baselines are not provided in pymarl2, they are using similar techniques introduced in [1] and pymarl2, e.g., FOP uses td-lambda, etc. Nevertheless, we are trying our best to promote a fair comparison in this paper.
>
> We appreciate your raising awareness of the need for a fair comparison here, which is what we also aim to do; and thus the hyperparameters are not fine-tuned in different experiments and the baselines are chosen from their source codes with their hyperparameters. This allows us to confirm the improvements are coming from our new designs rather than non-algorithm-related training tricks. More explanations on this and proper citations have been updated in the revised version.

---

> > ### Author Response · Authors · 2022-08-02
> > **Response to Reviewer cYxS (cont'd )**
> >
> > > Reviewer: Weaknesses： The authors claim that their method is SOTA, but the performance of the experiment is puzzling, for example, the performance on smac environment does not look good. Are there some differences in code implementation details.
> >
> > * Response: As we mentioned before, the results of QMIX and WQMIX are acquired from a fine-tuned implementation from pymarl2 (https://github.com/hijkzzz/pymarl2), and other baselines are chosen from their source codes with their hyperparameters, while the hyperparameters of all algorithms (including our proposed PAC) are not fine-tuned in different maps and experiments. In this way, we can establish a fair comparison and attribute the improvements to our new design rather than code-level optimization, e.g. hyper-parameter tuning. But still, we show overall better performance overall selected baselinse, some by a large margin on SMAC difficult maps.
> >
> >
> >
> > [1] Hu, Jian, et al. "Rethinking the implementation tricks and monotonicity constraint in cooperative multi-agent reinforcement learning." arXiv e-prints (2021): arXiv-2102.

---

> > > ### Comment · Reviewer_cYxS · 2022-08-03
> > > **Reply To Authors**
> > >
> > > Dear Authors.
> > >
> > > Thank you for your detailed replies, which solved part of my doubts.
> > >
> > > However, I still insist that hyperparameters should be adjusted when performing ablation experiments to obtain more accurate results. It is worth noting that the optimal hyperparameters obtained when we perform hyperparameter search for method A do not ensure that it is the optimal hyperparameters on method B. This means that when method B works better than method A, we can be sure that method B is superior to method A, but when method A is superior to method B, we cannot be sure whether it is the advantage of the method itself or the problem caused by the hyperparameters.
> > >
> > > Meanwhile, I noticed that in the experiments on SMAC, different MAPs used different training steps, and this step was much smaller than the number of steps used in pymarl2. So I am confused why the authors used different steps on different MAPs, and also whether the methods have converged. Would more training lead to better results for the baseline?

---

> > > > ### Author Response · Authors · 2022-08-08
> > > > **Response to Reviewer reply**
> > > >
> > > > We thank the reviewer for the comments and are glad that our previous response solved partial doubts of yours.
> > > >
> > > > * We now understand the concerns of the reviewer regarding the ablation studies design and performed hyperparameter-tuning for different versions of the algorithm in our ablation study. The new results are updated in the revised ablation studies section. To be specific, we did a grid search based hyperparameter-tuning on three key hyperparameters, regarding exploration steps ($\epsilon$ in $\epsilon$ - greedy, epsilon_anneal_time=[50k, 100k]), eligibility traces ($\lambda$ =[0.3, 0.5, 0.6] in TD-lambda), and replay-buffer size (buffer = [5000, 10000]). As expected, we find that different algorithms’ performances in our ablation study are affected by the hyperparameter tuning, while our main observations still remain valid. We now update the ablation studies section results with the hyperparameter-tuned versions and have details explained in the appendix.
> > > >
> > > > *  Although the steps selected for training are smaller than the steps used in pymarl2, we note that pymarl2 is a research focused on investigating the influence of code-level optimization and may require longer training steps to demonstrate the difference between hyperparameter selections. It is true that for some SMAC maps we have the training steps where some methods seem not converged yet. In fact, we are following the training steps settings that are no shorter than the steps in the state-of-the-art works we explained in related work, which should be considered enough training steps. To be specific, a comparison chart summarizing the choice of training steps is provided as below:
> > > >
> > > > |     Map    | Our Steps |       SOTA Steps      |
> > > > |:----------:|:---------:|:---------------------:|
> > > > |  5m_vs_6m  |     4M    |    QPLEX, WQMIX:2M    |
> > > > |    MMM2    |     2M    | FOP, FACMAC,QPLEX: 2M |
> > > > | 27m_vs_30m |     2M    |       FACMAC: 2M      |
> > > > |  6h_vs_8z  |     5M    |    WQMIX, LICA: 5M    |
> > > > |  corridor  |     5M    |   WQMIX:5M QPLEX:2M   |
> > > > |  3s_vs_5z  |     3M    |       QPLEX: 2M       |

---

> > > > ### Author Response · Authors · 2022-08-09
> > > > **Response to Reviewer**
> > > >
> > > > Thank you again for your review and the reply. We believe we’ve covered most of your concerns in the revised version and rebuttal now.  Please consider raising our score, or let us know if there are other places that need addressing. Thank you!

---

### Official Review · Reviewer_wLBP · 2022-07-12

**Rating:** 6
**Confidence:** 4
**Soundness:** 3 good
**Presentation:** 3 good
**Contribution:** 3 good

**Summary:**

This paper aims at addressing the estimation error caused by partial observation in value decomposition methods. A new framework leverages assistive information generated from counterfactual predictions of optimal joint action selection, which guides the Q values of each agent. Experiments show that the proposed method overperforms many baselines including QMIX.

**Questions:**

1. Authors mention QTRAN in the paper but do not compare it in the experiments and Figure 1. Is there any reason?
2. One existing work [1] is missing, which is the state-of-art method in this domain. I think it can also address the problem shown in Figure 1 since it has complete expressiveness.

[1] QPLEX: Duplex Dueling Multi-Agent Q-Learning. Wang et al. 2020

**Limitations:**

null

**Strengths And Weaknesses:**

Strengths: 1. novel method to use counterfactual loss in value decomposition methods. 2. Experiments based on StarCraft II show the improvement of the proposed method. 3. Good writing.
Weaknesses: 1. Some existing works are not compared.

---

> ### Author Response · Authors · 2022-08-02
> **Response to Reviewer wLBP**
>
> Thank you for your comments and for confirming the contributions of our paper. We provide clarification to your questions and concerns as below.
> > Reviewer: Authors mention QTRAN in the paper but do not compare it in the experiments and Figure 1. Is there any reason?
> * Response:  We very much appreciate the idea and design of QTRAN. However, later research points out that QTRAN in fact performs poorly in empirical testing environments due to several bottleneck issues [3, 4]. QTRAN is also shown to be outperformed by one of the later works that we use as our baselines DOP [2]. Since we have compared with a more recent baseline DOP that is known to perform better than QTRAN in these environments, we do not explicitly compare with it. We will add an explanation in the revision that QTRAN is not compared since the authors of [2] showed that DOP achieves better results than QTRAN and thus we compare our approach with DOP.
>
> > One existing work [1] is missing, which is the state-of-art method in this domain. I think it can also address the problem shown in Figure 1 since it has complete expressiveness.
> * Response: While QPLEX [5] achieves a higher expressiveness of value function factorizations by introducing duplex dueling architecture, a later work FOP [6] – which we have selected as a baseline in the experiment results –  shows a higher performance than QPLEX in almost all scenarios. In the evaluation, we have decided to use the more recent FOP as a baseline. Our algorithm is demonstrated to outperform the state-of-the-art method FOP, which is known to outperform QPLEX (as shown in [6]).  We add this explanation in the revised version.
>
> > Reviewer: Weaknesses: Some existing works are not compared
> * Response: As mentioned above, we have decided to use more recent baselines (i.e., FOP and DOP) that are known to outperform QTRAN and QPLEX in the evaluation. In general, we tend to choose baselines that are more closely related to our work and most recent. This motivated the choice of QMIX (baseline for value-based factorization methods), WQMIX (close to our work that uses weighted projections so better joint actions can be emphasized), NDQ [8] (which similarly uses common information to assist decision making but as generating messages for agent-wise communication), VDAC [7], FOP [6], DOP [2] (SOTA actor-critic based methods). These approaches have been shown to outperform the other approaches mentioned by the reviewer. In total, we have 6 other algorithms as baselines for comparison. We believe our choice of baselines is sufficient, and we updated the reasons for baseline selection in the revised version.
>
>
>
> [2] Wang, Yihan, et al. "Dop: Off-policy multi-agent decomposed policy gradients." International Conference on Learning Representations. 2020.
>
> [3]Son, Kyunghwan, et al. "QTRAN++: improved value transformation for cooperative multi-agent reinforcement learning." arXiv preprint arXiv:2006.12010 (2020).
>
> [4]Mahajan, Anuj, et al. "Maven: Multi-agent variational exploration." Advances in Neural Information Processing Systems 32 (2019).
>
> [5] Wang, Jianhao, et al. "Qplex: Duplex dueling multi-agent q-learning." arXiv preprint arXiv:2008.01062 (2020).
>
> [6] Zhang, Tianhao, et al. "Fop: Factorizing optimal joint policy of maximum-entropy multi-agent reinforcement learning." International Conference on Machine Learning. PMLR, 2021.
>
> [7] Su, Jianyu, Stephen Adams, and Peter Beling. "Value-decomposition multi-agent actor-critics." Proceedings of the AAAI Conference on Artificial Intelligence. Vol. 35. No. 13. 2021.
>
> [8] Wang, Tonghan, et al. "Learning nearly decomposable value functions via communication minimization." arXiv preprint arXiv:1910.05366 (2019).

---

> ### Author Response · Authors · 2022-08-08
> **Response to Reviewer wLBP**
>
> Thank you for your review. In the revision, we have provided the additional baseline results. In specific, we added the SMAC performance of QPLEX in the main results and QTRAN multi-state matrix game result in the Appendix.

---

### Meta-Review · Area_Chair_yqty · 2022-08-25

**Recommendation:** Accept
**Confidence:** Less certain

**Metareview:**

After reading the reviews and feedbacks, I lean towards acceptance. A majority of reviewers gave a positive score after the rebuttal period and some concern were answered in the authors response. Specifically authors have shown that the recent baselines they use outperform other baselines and therefore those do not need to be added in the paper, they have also clarified some proofs and some notations. Overall, the reviewers found the method presented interesting, the paper well written and appreciated the comparison to other methods of the literature. Finally, experiments shows interesting results on large scale domains which is a sign that the proposed method could scale up.

**Award:**

No

---

### Decision · Program_Chairs · 2022-09-14

Accept